# Spatial microenvironments tune immune response dynamics in the *Drosophila* larval fat body

**Brandon H. Schlomann**[1,2]*, **Ting-Wei Pai**[1], **Jazmin Sandhu**[2], **Genesis Ferrer Imbert**[1,2], **Thomas G. W. Graham**[1], **Hernan G. Garcia**[1,2,3,4,5,6]

**1** Department of Molecular and Cell Biology, University of California, Berkeley, California, United States of America, **2** Department of Physics, University of California, Berkeley, California, United States of America, **3** Institute for Quantitative Biosciences-QB3, University of California, Berkeley, California, United States of America, **4** Chan Zuckerberg Biohub – San Francisco, San Francisco, California, United States of America, **5** Biophysics Graduate Group, University of California, Berkeley, California, United States of America, **6** Graduate Program in Bioengineering, University of California, Berkeley, California, United States of America

\* bschloma@berkeley.edu

## Abstract

Immune responses in tissues display complex spatial patterns of gene expression that are linked to disease outcomes. However, the processes that generate these patterns—including the relative roles of noisy gene expression dynamics, microbial transport, and tissue anatomy—are poorly understood. As a tractable model of spatial immune responses, we investigated heterogeneous expression of antimicrobial peptides in the larval fly fat body, an organ functionally analogous to the liver. To quantify single-cell antimicrobial peptide expression dynamics in the fat body, we developed a protocol for light sheet fluorescence microscopy of whole, live larvae. Using this approach, we discovered that individual fat body cells express antimicrobial peptides at approximately constant rates following infection, but that the average rate varies along the anterior-posterior axis of the fat body, with rapid expression in the anterior and posterior lobes. Overexpression of immune signaling components and analysis of spatial transcriptomes revealed that these tissue microenvironments are predefined independently of infection, with the rate-limiting step of antimicrobial peptide induction downstream of peptidoglycan sensing. The locations of these microenvironments correlate with heartbeat-dependent fluid flow in a manner resembling the strategic positioning of immune cells in the liver, gut, and lymph nodes of mammals. We speculate that this spatial compartmentalization helps the fat body efficiently perform its diverse metabolic, enzymatic, and immunological functions.

**Data availability statement:** The raw and processed image data from this manuscript can be obtained from a public AWS S3 bucket with URL: https://s3.console.aws.amazon.com/s3/buckets/schlomann-diptericin-paper or URI: //s3://schlomann-diptericin-paper/Data/. The spatial transcriptomics data was downloaded from the Flysta3d homepage, https://db.cngb.org/stomics/flysta3d/download/. Tables of identified marker genes and data extracted from images are included with the manuscript as Supplemental Data Files. Code for reproducing the plots of the paper is found at https://github.com/bschloma/diptericin-paper.

**Funding:** B.H.S. was supported by a James S. McDonnell Foundation fellowship, award number 2020-1172. T.G.W.G. was supported by the Jane Coffin Childs Memorial Fund for Medical Research. H.G.G. was supported by NIH R01 Awards R01GM139913 and R01GM152815, by the Koret-UC Berkeley-Tel Aviv University Initiative in Computational Biology and Bioinformatics, and by a Winkler Scholar Faculty Award. H.G.G. is also a Chan Zuckerberg Biohub Investigator (Biohub – San Francisco). The funders had no role in study design, data collection and analysis, decision to publish, or preparation of the manuscript.

**Competing interests:** The authors have declared that no competing interests exist.

## Author summary

Recent sequencing and imaging technologies have revealed that immune responses in our organs are not spatially uniform, but occur in complex patterns in which clusters of nearby cells are strongly active. There is increasing evidence that these spatial interactions are important for controlling disease outcomes. However, little is known about the dynamics of how these spatial patterns form: are they created through randomness, are they shaped by external signals, such as pathogen localization, or are they predetermined, representing a fine-grained tissue anatomy? While it is practically infeasible to directly observe these types of cellular dynamics in humans or even mice, small, transparent organisms like fruit fly larvae offer a literal window into the inner workings of immune responses. We used a combined imaging and genetics approach to study heterogeneous spatial patterns of antimicrobial peptide production in the fruit fly equivalent of the liver. We discovered that these spatial patterns were in fact predetermined and represent previously unknown immune microenvironments within this important tissue that correlate with areas of fast blood flow. Since innate immune signaling in highly conserved, this spatial logic may be a general feature of immunological tissues that is relevant to other animals, including humans.

## Introduction

Immune responses in tissues exhibit complex spatiotemporal patterns of gene expression and cellular behaviors [1–4]. Recent advances in our ability to map immune responses in space at single-cell resolution have identified gene expression patterns that correlate with disease severity in infections [5,6] and cancer [7,8]. Several mechanisms are known for generating heterogeneous immune responses, including stochastic transcription [9–14], amplification steps in immune signaling pathways [15], cell-cell communication via secreted cytokines [16], the anatomical structure of tissues [1], and the spatial distribution and behavior of microbes [17,18]. However, understanding how all of these processes, which act at different length and time scales, combine to generate observed spatial patterns of immune response remains an open fundamental problem.

The fruit fly is a well-established model system for understanding innate immunity [19,20]. Flies possess highly conserved immune signaling pathways, including those involving NF-$\kappa$B family transcription factors, which activate a variety of immune effectors [19,20]. The fly's primary defense against systemic bacterial infection is the secretion of antimicrobial peptides from an organ called the fat body, which serves similar functions to those of the mammalian liver and adipose tissue combined [19,20]. Fluorescent reporters of antimicrobial peptide production can be used to read out immune activity and are particularly useful in fly larvae, which are more optically transparent than adult flies [21]. Structurally, the larval fat body consists of multiple monolayer sheets of cells suspended within the internal fluid of the insect, called the hemolymph. Previous observations indicated that expression of one antimicrobial peptide, Diptericin-A (DptA), can exhibit a spatially heterogeneous expression pattern

within the fat body, with random variation on the scale of a few cells [22,23] and gradients of activation along the anterior-posterior axis [24]. However, the mechanisms behind this pattern, the dynamics of its formation, and the extent of its generality across immune response components and stimuli were unclear.

One hypothesis for the mechanism behind the spatial gradients in immune response is that the fat body is divided into distinct compartments that are intrinsically heterogeneous. While the fly fat body is thought to comprise a single cell type, different lobes of the larval tissue arise from distinct clusters of progenitor cells in the embryo [25], so it is plausible that they acquire heterogeneity during development. Classic works observed distinct enzymatic activity [26,27], protein granule formation [28], and tumorogenesis [29] in different spatial regions of the fat body. More recently, factors involved with lipid metabolism [30] and anti-parasitoid immunity [31] were found to vary in fat body cells along the anterior-posterior axis of larvae. Within adult flies, single-nucleus RNA sequencing recently revealed widespread heterogeneity among fat body cells [32,33]. More broadly, outside of *Drosophila*, functional differentiation within the fat bodies of other insects has been observed and is thought to be an important aspect of this highly multifunctional tissue [34].

An alternative hypothesis is that spatial structure in antimicrobial peptide expression is driven by spatial structure in microbial stimuli. Recent fluid mechanical studies have found that the insect hemolymph is capable of compartmentalization and gradient formation [35]. Further, microbes can use motility and aggregation to control their own spatial organization within complex fluid environments, which can result in spatially-restricted immune responses [18].

In this study, we tested these two hypotheses with a combined genetics and imaging approach. To enable quantitative measurements of antimicrobial peptide expression heterogeneity in space and time within live animals, we developed a protocol for imaging whole larvae with light sheet fluorescence microscopy [36,37]. With this approach, fluorescent reporters of gene expression [21], and computational image analysis, we quantitatively measured immune response and bacterial dynamics across whole larvae ($\approx$3mm long) with single-cell resolution during systemic infections. Our observations, together with genetic perturbations and analysis of spatial transcriptomes [38], firmly establish the existence of spatial microenvironments within the larval fat body that are primed for fast rates of antimicrobial peptide production in a manner independent of infection. While more work is needed to identify the molecular mechanisms defining these microenvironments, we identified that within the immune deficiency (Imd) pathway—the primary signaling pathway that mediates the DptA response—the mechanisms acts downstream of peptidoglycan receptor activation, with some evidence suggesting that it acts downstream of nuclear import of the key NF-$\kappa$B transcription factor, Relish. Our findings extend the scope of spatial compartmentalization within the insect fat body [29–31,34] and more generally support the notion of strategic cellular positioning [1] within immunological tissues as a conserved design principle of immune systems.

## Results

### DptA and other antimicrobial peptides are expressed in a robust spatial pattern within the fat body

To study mechanisms driving cell-cell variability in DptA expression within the fat body, we developed an infection protocol that produces DptA responses that are spatially heterogeneous yet reproducible, with 100% of larvae containing some amount of detectable signal from a well-established DptA-GFP transcriptional reporter [21] (S1 Fig). Our protocol uses precisely staged early third instar (L3) larvae, which have reduced immune reactivity due to lower levels of the steroid hormone Ecdysone [22,23,39–42]. Infection is achieved through microinjection with specialized, fine-tipped quartz glass needles that enable the delivery of high bacterial loads while minimizing wounding (*Methods: Microinjection*). We checked that DptA-GFP levels represent a linear, quantitative measure of gene expression by measuring that animals homozygous for the reporter contain a median fluorescence intensity ($6 \pm 4 \cdot 10^6$ a.u.), approximately twice that of heterozygous animals ($2.7 \pm 0.9 \cdot 10^6$ a.u.; S2A Fig). We also confirmed that the ether anesthetic used to immobilize larvae for injections does not affect DptA-GFP levels by comparing to larvae immoblized by cold shock (S2B Fig).

With our infection protocol established, we began by measuring total DptA-GFP fluorescence intensity 24 hours after infection, long after the initial activation of DptA, which occurs between 3-5 hours post infection [22]. Due to the high stability of GFP in vivo [43], this measurement is a proxy for the total amount of DptA produced over the course of the infection. Both mock injected and non-injected larvae produced zero observable DptA-GFP signal (Fig 1A, gray circles). Comparing larvae injected at different times within early third instar (3 hours post molt to third instar at 25°C, 18 hours at 18°C, and 18 hours at 25°C), we observed a monotonic increase of DptA-GFP levels with developmental stage (Fig 1A, green circles). We found that injecting larvae at the 18h-18°C time point produced responses with the strongest within-fat body heterogeneity and chose to characterize this stage further.

Using image analysis (*Methods: Image analysis*), we quantified DptA-GFP fluorescence intensities within individual cells. Larvae cleanly clustered into two populations based on median cell intensity (Fig 1B), which we denote as "complete response" and "partial response". These two clusters did not correlate with fat body length, which is a proxy for developmental stage and thus Ecdysone levels, or experiment date (S3A Fig). We did observe a partial correlation with larva sex, with 6/6 male larvae exhibiting a partial response, which may be indicative of X-linked Ecdysone effects [22], though female larvae were split evenly across partial and complete responses (6 partial, 8 complete). Since larvae of both sexes exhibited examples of partial responses, we continued to analyze both males and females in all experiments. Complete responses were uniform across the fat body (Fig 1C,ii). In contrast, partial responses were highly heterogeneous (Fig 1C,i and S1 Movie); mock-injected larvae showed no detectable expression (Fig 1C,iii).

Strikingly, cellular variability in DptA expression had a highly stereotyped spatial pattern: mean expression was high in the anterior and posterior fat body, but lower in the middle (Fig 1D; the plot shows mean and standard deviation across $N = 12$ larvae, with expression profiles along the anterior-posterior axis normalized to the maximum level to account for variability in the overall level of DptA expression). The balance of expression between anterior and posterior fat body varied between larvae: some larvae had stronger expression in the anterior than in the posterior, some had the reverse, but one or both of anterior and posterior always had between 2 and 10 times higher mean expression than the middle (S3B Fig). This spatial pattern of DptA expression was independent of injection site on the larva (S2C-S2F Fig). Quantitative inspection of expression variability at the single-cell level revealed that partial responses exhibit a broad, continuous distribution of expression levels that ranges from zero detectable expression (consistent with mock injections) all the way up to levels consistent with complete responses (Figs 1E and S3C).

This robust spatial pattern of DptA expression observed following *E. coli* microinjection is consistent with previous observations during oral infection with *Erwinia carotovora carotovora 15* (*Ecc15*) [24]. We also observed this spatial pattern of DptA expression following microinjection *Providencia rettgeri* strain Dmel (S4A-S4C Fig). These results suggest that this spatial pattern of DptA expression is a general feature of immune activation in early L3 larvae.

To further assess the generality of this response, we next examined whether this spatial expression pattern was unique to DptA, or was shared among other antimicrobial peptides. We screened a suite of 6 additional reporter constructs [21] that span the full family of classical antimicrobial peptides in fruit flies: Attacin, Cecropin, Defensin, Drosocin, Drosomycin, and Metchnikowin. Specifically, we asked whether larvae with partial responses, in which only a subset of fat body cells express the reporter, exhibited a similar "U-shaped" expression pattern to DptA. The first 4 antimicrobial peptides are known to be downstream of the Immune deficiency (Imd) pathway in the fat body [44] (Fig 2A), which senses DAP-type peptidoglycan, and so were induced using the same *E. coli* infection protocol as DptA. The last two peptides are known to be induced in the fat body primarily by the Toll pathway [44], which senses Lys-type peptidoglycan and fungal $\beta$-glucan (Fig 2A), and so were induced by microinjection with yeast, *S. cerevisiae* or the Gram positive bacterium *E. faecalis*.

We found that antimicrobial peptide responses varied considerably both within and across peptides (Fig 2B). Attacin-A and Drosomycin were strongly expressed in all fat body cells in all larvae, and so we were unable to assess the spatial patterning of partial responses for these genes. In contrast, Cecropin-A1 was barely detectable, with only a small number of cells in a small number of larvae positive for GFP, preventing robust assessment of spatial patterning of this gene.

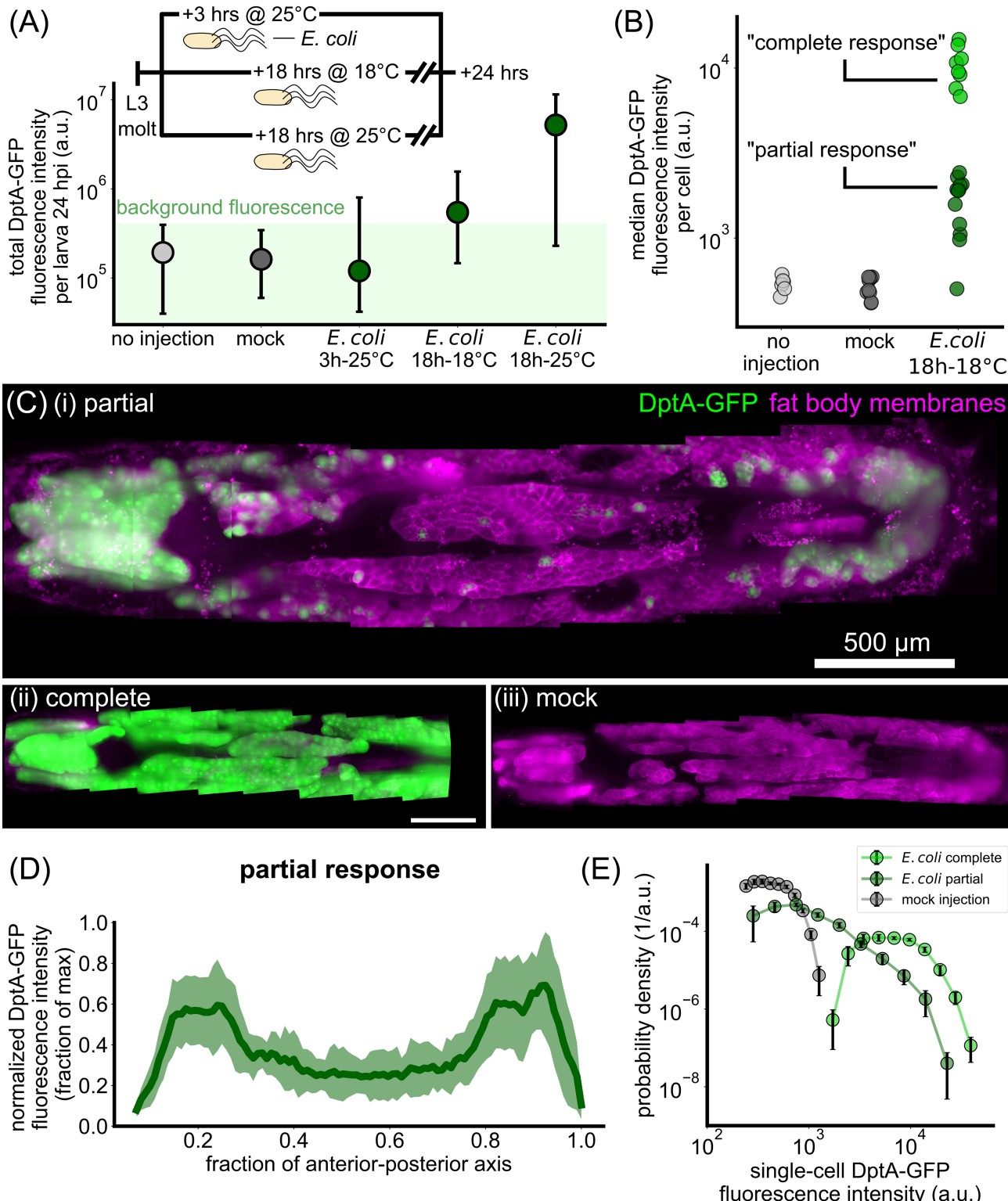

**Fig 1**. **The antimicrobial peptide reporter DptA-GFP is expressed heterogeneously throughout the fat body but exhibits a reproducible spatial pattern along the anterior-posterior axis during early third instar.** (A) The inducibility of DptA increases with larval age. Total fluorescence intensity of DptA-GFP per larva at 24 hours after infection (hpi) with *E. coli* is plotted as a function of age after molt to L3. Inset shows the experimental timeline. Circles denote median values, bars denote quartiles. Age is denoted by hours after molt to L3 at a given temperature in degrees Celsius. Larvae

aged 18 hours post L3 molt at 18°C at the time of infection produce intermediate DptA expression levels, and are the focal age of the paper. Injection and mock groups showed no detectable DptA-GFP signal and thus represent the measured range of background fluorescence. (B) From image-based quantification of single-cell DptA-GFP levels, we plot the median single-cell expression level for each larva and find that larvae cluster into two groups, denoted "partial responses" and "complete responses". (C) Maximum intensity projections of larvae showing DptA-GFP (green) and fat body membranes (magenta, UAS-mCD8-mCherry; r4-Gal4). In all images, anterior is to the left. A representative partial response (i) exhibits high expression in the anterior- and posterior-dorsal fat body, with minimal, scattered expression in the middle fat body. Complete responses (ii) exhibit a uniform expression pattern, while mock injected larvae (iii) show no detectable expression. Timing is 24 hours after injection. DptA-GFP channel is log-transformed and all images are adjusted to the same contrast levels. Scale bar in (ii) is 500 $\mu$m. (D) Quantification of the "U-shaped" DptA-GFP expression pattern for partial responses only. Each larva's expression pattern is normalized to its maximum value and then averaged (green line). Shaded error bars denote standard deviation across $N = 12$ larvae. (E) Probability densities of single-cell DptA-GFP expression levels for mock (gray), partial responses (dark green), and complete responses (bright green), showing that partial responses comprise a continuous, broad distribution of expression levels.

However, Drosocin and Defensin exhibited clear examples of a "U-shaped" partial response, mirroring DptA (Fig 2C-2D). Metchnikowin exhibited strong expression only in the anterior, not posterior fat body following injection with *S. cerevisiae* (Fig 2E). However, both Metchnikowin and Drosomycin showed uniform expression patterns following injection with *E. faecalis* (S4D-S4I Fig). Together, these data indicate that spatial patterning of immune response in the larval fat body—particularly enhanced expression in the anterior-dorsal lobes—is not restricted to DptA, but is a more general phenomenon of the Imd pathway. We find mixed evidence of spatially patterning in the Toll pathway, and so restrict our analysis to the Imd pathway for the remainder of the paper.

With this repeatable yet heterogeneous immune response expression pattern characterized, we next sought to understand the origins of both cell-cell variability within fat body regions and the overall spatial patterning across the tissue. We began by leveraging our live imaging capabilities to characterize the dynamics of immune response pattern formation.

### Single-cell DptA-GFP expression dynamics are deterministic with spatially-varying rates

Variability in DptA-GFP levels 24 hours after infection could arise from multiple different types of dynamics. The highest expressing cells could have the highest rates of DptA expression, the shortest delay before beginning to respond, the largest fluctuations as part of a highly stochastic response, or a combination thereof. To distinguish between these dynamical modes of activation, we adapted our light sheet fluorescence microscopy mounting protocol to enable continuous imaging of live larvae for several hours (*Methods: Light sheet fluorescence microscopy*) and obtained movies of two larvae exhibiting partial responses.

Levels of DptA-GFP visibly increased over the course of the movies, with a clear bias of expression in the anterior-dorsal lobes of the fat body (Fig 3A, S2 and S3 Movies). Using image analysis (*Methods: Image analysis*), we quantified the dynamics of expression in 227 cells across 2 movies and pooled the data for analysis (Fig 3B). While each single-cell measurement contained substantial noise due to fluctuating background levels and tissue motion, the overall trends were smooth increases in DptA levels in all cells, with spatially varying rates. Fitting a linear rise to the initial phase of activation, we found that single-cell activation rates in the anterior fat body are uniformly high compared to rest of the tissue, with a median rate roughly twice that of the middle region (Fig 3C). The middle region contains more variability, with a continuous spread of rates ranging from zero expression all the way to rates consistent with the anterior (Fig 3B–3C). The posterior region has a 60% higher median rate than the middle, though there is also a wide, continuous spread in the rates. Finally, there is a moderate correlation ($R^2 = 0.61$) between the initial activation rate and the level of DptA expression at 6 hours post infection (Fig 3D), suggesting that expression rate, not delay, primarily determines long-term expression level. Due to the high stability of GFP, our measurements are insensitive to potential high frequency fluctuations in DptA expression. However, overall, the data support a model of largely deterministic expression with spatially varying rates, rather than one of varying activation delays or strongly stochastic dynamics.

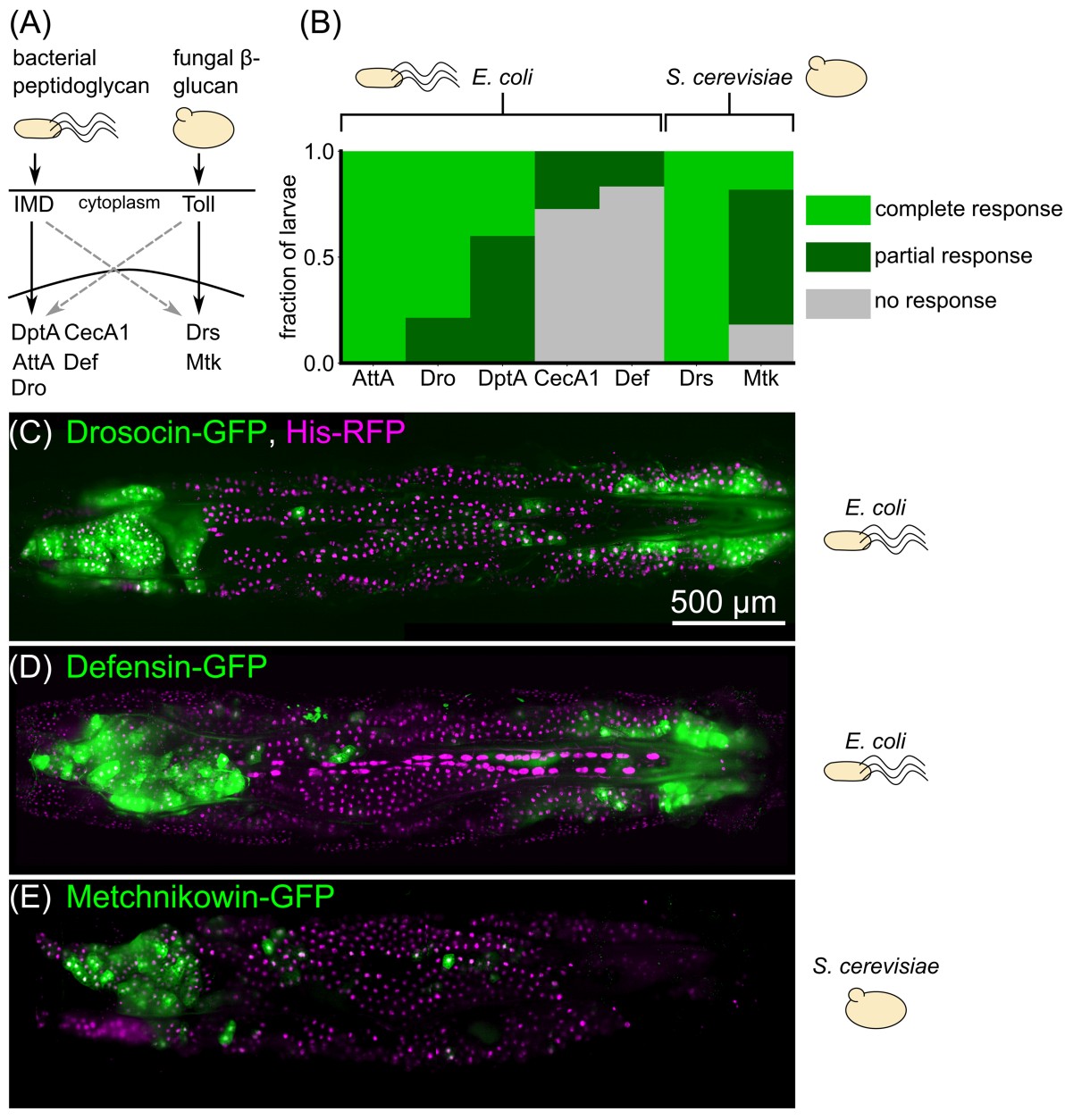

**Fig 2. Spatial patterns of expression upon immune challenge occur in a variety of antimicrobial peptides.** (A) Simplified schematic of the main immune signaling pathways in *Drosophila*. Bacterial peptidoglycan is sensed through the Immune deficiency (Imd) pathway, which leads to activation of Diptericins (including DptA), Cecropins (including CecA1), Attacins (including AttA), Defensin (Def), and Drosocin (Dro). Fungal $\beta$-glucan is sensed through the Toll pathway and leads to activation of Drosomycin (Drs) and Metchnikowin (Mtk). There is cross-talk between the pathways (dashed gray arrows). (B) Fraction of larvae exhibiting partial (subset of fat body cells GFP⁺), complete (all fat body cells GFP⁺), or no response of GFP-reporters of various antimicrobial peptides following challenge with *E. coli* or *S. cerevisiae*. Responses were scored based on images taken on a low-magnification widefield microscope 24 hours post infection, except for DptA, which were taken from the light sheet fluorescence microscopy data from Fig 1. All larvae were staged to 18h post-L3 molt at 18°C (*Methods: Larva collection*). Sample sizes (number of larvae) for each gene, left to right: $N$ = 7, 14, 20, 11, 12, 8, 11. (C)-(E) Maximum intensity projections of light sheet fluorescence microscopy image stacks of larvae carrying GFP reporters for Drosocin (C), Defensin (D), and Metchnikowin (E), with the microbial stimulus used noted to the right of each image. Fat body nuclei are marked using UAS-His-RFP; r4-Gal4. Image contrast was adjusted for each panel separately for visual clarity.

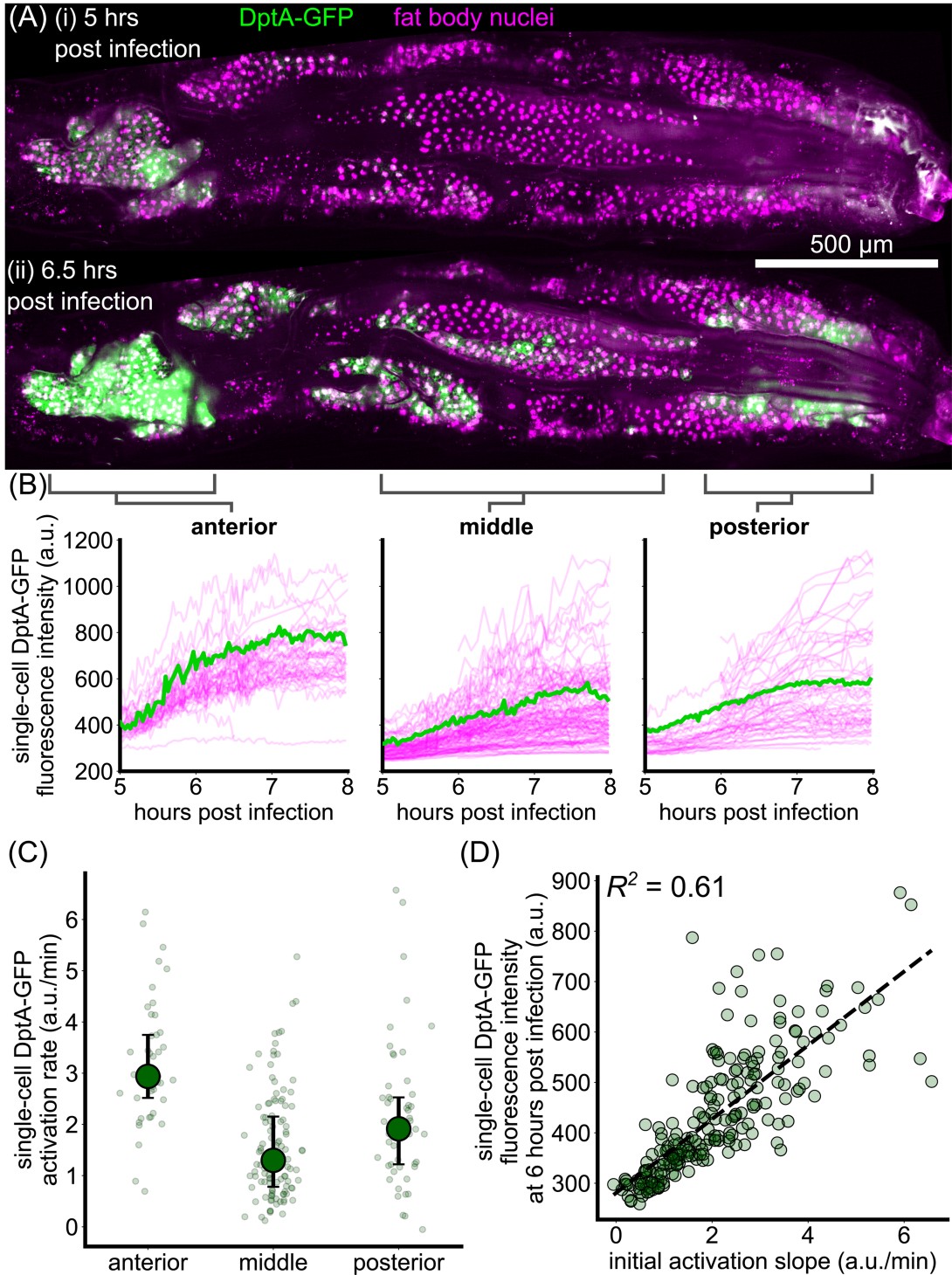

**Fig 3. Single-cell dynamics of DptA expression exhibit smooth activation with spatially-varying rates.** (A) Maximum intensity projection snapshots of DptA-GFP expression during time-lapse imaging. Time denotes hours post infection. The images come from S2 Movie. See also S3 Movie. (B) Single-cell traces of mean DptA-GFP expression per cell over time from cells in 3 regions of the dorsal fat body. One representative trace from each region is highlighted in green, the rest are drawn in magenta. The data are pooled from movies of $N = 2$ larvae (S2 and S3 Movies). (C) Single-cell DptA activation rates, defined by linear fits to the initial rise in DptA-GFP traces in (B), in anterior, middle, and posterior regions of the fat body. Large circles and error bars denote quartiles. Small circles represent individual cells. (D) Instantaneous fluorescence intensity 6 hours post infection strongly correlates with the initial rate of production. Each marker is a cell.

We validated our light sheet fluorescence microscopy-based measurements by manually following individual larvae for several hours on a widefield microscope (*Methods: Widefield microscopy*, S5 Fig). While this approach does not allow segmentation of single cells due to strong background fluorescence, we were able to quantify tissue-scale activation dynamics in the anterior fat body. We observed a similar pattern of smooth increase in expression that resulted in signal that is approximately twice as bright as background levels by 6 hours post infection.

Having inferred that DptA patterning is due primarily to a deterministic modulation of expression rate along the anterior-posterior axis, we searched for the drivers of this variability, beginning with bacterial stimuli.

## Heartbeat-induced hemolymph flow correlates with, but does not cause, DptA patterning

DptA is activated by the Imd pathway, which in turn is activated by the binding of peptidoglycan to a membrane-bound receptor [19]. We hypothesized that the observed spatial pattern of DptA expression might be caused by spatial localization of peptidoglycan or bacteria themselves via hemolymph flow. The larval circulatory system consists of a single tube suspended in the hemolymph that pumps peristaltically from the posterior to the anterior at a frequency of around 4 Hz [45]. Dye injected into the posterior of larvae is rapidly transported through the heart and deposited in the anterior (Fig 4A–Fig 4B), just beneath the anterior lobes of the fat body that display enhanced immune activity. This pattern of flow was dependent on the heartbeat and was eliminated following loss of heartbeat by heart-specific knock down of myosin heavy chain (Mhc) using NP1029-Gal4 driving UAS-Mhc-RNAi (S4 Movie), following reference [46] (Fig 4C–Fig 4D). (Note that heartbeats are not required for survival in flies [45], as flies acquire oxygen directly through the air via their tracheal system—a consequence of their small size and the fast diffusion rate of oxygen in air.) These observations indicate that heartbeat-induced fluid flow—which may carry immunogenic compounds—correlates with the spatial pattern of antimicrobial peptide production.

We next investigated how bacteria were distributed within the hemolymph. Fluorescent *E. coli* were directly transported through the heart (known as the "dorsal vessel"), flowing from the posterior to the anterior at a speed of approximately 1 mm/s (S5 Movie). However, the density of planktonic bacteria in the hemolymph 6 hours post infection only showed a mild "U-shaped" pattern along the anterior-posterior axis (Fig 4E–Fig 4F; see also  and S7A Figs, S6 Movie, and *Methods: Bacteria segmentation* for quantification details); loss of heartbeat shifted the distribution slightly to the posterior and increased the overall number of bacteria at this time point (Fig 4G–Fig 4H). As with the rhodamine dye, the flow of bacteria through the heart was also eliminated following loss of heartbeat, though additional fluid flows were still present due to body wall contractions (S7 Movie). Plotting DptA-GFP fluorescence intensity against planktonic bacterial density in the same anterior-posterior axis bins revealed distinct input-output relationships in the anterior and posterior regions, suggesting that average bacterial concentration is not the sole determinant of DptA expression (S7B Fig).

To directly test the role of the heartbeat in the spatial pattern of antimicrobial peptide expression, we combined our DptA-GFP reporter with the heart-specific myosin knockdown and assessed DptA levels 6 hours post infection (recall that DptA levels at this time point correlate with activation rates at the single cell level; *Methods: Heartbeat knockdown experiments*; S8 Fig). Control larvae containing only the reporter and UAS-Mhc-RNAi showed the expected "U-shaped" expression pattern along the anterior-posterior axis (Fig 4I–Fig 4J). In contradiction to our hypothesis, larvae lacking a heartbeat also showed a strong "U-shaped" expression pattern, indicating that the heartbeat is not required for spatially-patterned DptA expression (Fig 4K–Fig 4L).

In addition to knocking down myosin, we eliminated the heartbeat by overexpressing the potassium channel Ork1, following [45]. While this strategy robustly eliminated the heartbeat throughout the larval stage, unexpectedly, we found that after injection with either *E. coli* or a mock control, the heartbeat restarted within 3-6 hours (S8 Movie), preventing us from using this approach to assess the role of the heartbeat in DptA expression. We note that the spatial pattern of DptA expression was unchanged by Ork1 overexpression (S9 Fig).

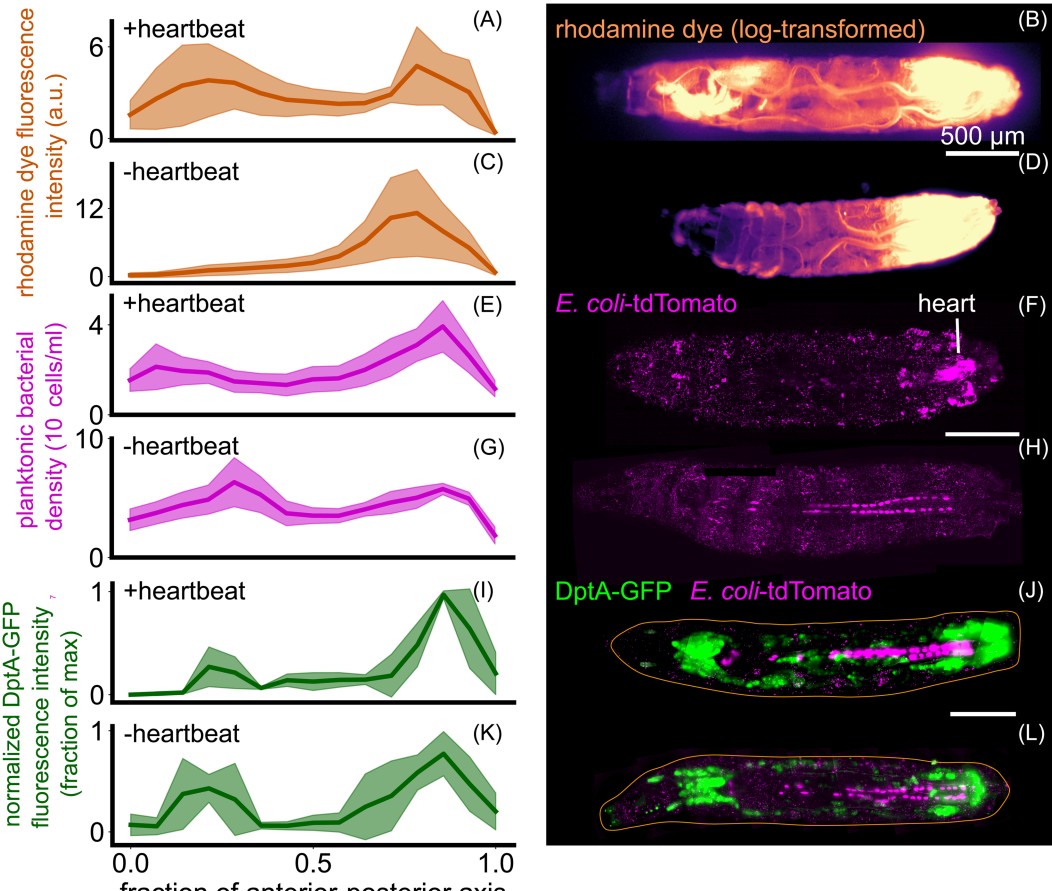

**Fig 4**. **Heartbeat-induced fluid flows mirror immune responses but are not required for patterning of DptA.** Each row shows quantification (left, mean and standard deviation) and a representative image (right) of various quantities in animals with and without heartbeats. The heartbeat was eliminated by myosin knockdown in the heart using NP1029-Gal4 driving UAS-Mhc-RNAi. (A)-(D) Rhodamine dye injected in the posterior and imaged 5 minutes after injection, with and without a heartbeat ($N = 5$ larvae per group). (E)-(H) *E. coli* 3 hours post injection with and without a heartbeat ($N = 4$ larvae per group). In the quantification, to avoid counting fluorescence internalized by host cells, planktonic bacteria freely suspended in the hemolymph were computationally identified and only these cells were counted (*Methods: Bacteria segmentation*, S6 Movie). (I)-(L) DptA-GFP 6 hours post injection in animals with and without a heartbeat ($N = 5$ larvae per group). All scale bars are 500 $\mu$m. In (J) and (L), the approximate outline of the larva is marked as an orange line. Images in (B) and (D) are single-plane widefield images. Images in (F), (H) (J), and (L) are maximum intensity projections of 3D light sheet images stacks.

Altogether, these results indicate that the observed spatial patterning of antimicrobial peptide production within the fat body is not due to spatial structure in microbial stimuli. Rather, these results suggest that the regions of enhanced immune activity in the fat body represent persistent spatial microenvironments that are primed for antimicrobial peptide expression prior to the start of infection. Since our heartbeat knockdown was in effect from the beginning of embryogenesis, we could conclude that the heartbeat itself is not involved in the immune priming. Therefore, we next sought to identify factors that define these microenvironments and to explicitly demonstrate that the heterogeneity in antimicrobial peptide response was independent of infection.

**Spatially heterogeneous DptA responses occur during overexpression of Imd pathway components.**

We hypothesized that different baseline levels of immune receptors between fat body cells in different regions of the tissue could explain differences in their diverging responses to saturating levels of bacteria. We tested this hypothesis through an overexpression strategy. Bacteria in the fly are sensed by the immune system through peptidoglycan receptors. For bacteria such as *E. coli* that contain DAP-type peptidoglycan, the primary receptor relevant for systemic infections is peptidoglycan recognition protein LC (PGRP-LC), which is spliced into two receptor isoforms, PGRP-LC.x and PGRP-LC.a [49]. The former binds polymeric peptidoglycan, while the latter binds monomeric peptidoglycan as part of a heterodimer (Fig 5A). Both receptors lead to activation of Imd, which starts a signaling cascade resulting in the nuclear import of the NF-$\kappa$B transcription factor Relish and subsequent activation of antimicrobial peptide genes [47].

To test the hypothesis that PGRP-LC activation is the rate limiting step for DptA induction, and to simultaneously confirm that the shape of the DptA expression pattern is independent of the microbial stimulus, we used the fact that antimicrobial peptides can be artificially induced by overexpressing peptidoglycan receptors [50]. We reasoned that if peptidoglycan receptors were the rate-limiting step of DptA expression, such that lower receptor levels in the middle fat body resulted in weak DptA expression, then overexpression of the receptors would produce consistently uniform DptA expression patterns. Since hemocytes are known to regulate antimicrobial peptide expression [51], we expressed PGRPs simultaneously in both the fat body and hemocytes, using the *cg*-Gal4 driver, to best mimic a systemic infection with a spatially-uniform stimulus (Fig 5B and *Methods: Imd pathway overexpression experiments*).

Strikingly, overexpression of PGRP-LC.a in the absence of infection consistently produced a clear "U-shaped" DptA expression pattern that mirrors the "partial response" seen in *E. coli* infections (Fig 5C–5F). Overexpression of PGRP-LC.x in the absence of infection also produced an expression pattern that was biased to the anterior and posterior, but also included more examples of larvae with expression in the lateral fat body lobes, which mildly altered the quantification of the one-dimensional expression distributions 5G–5H). Moving one step down the signaling cascade from the receptors, overexpression of a constitutively active form of Imd, ImdCA [48], produced more uniform anterior-posterior expression patterns (Fig 5I). However, visual inspection of larvae revealed that this was due to consistent expression within the lateral fat body lobes; expression of DptA in the middle dorsal fat body lobes was consistently absent compared to the membrane marker (mCD8-mCherry) controls (Fig 5J–5L). The membrane marker control showed mostly uniform expression across the fat body, though we did detect 1.3x more expression in the anterior lobes compared to the middle on average, which may be indicative of small differences in global aspects of gene expression between these tissue regions (Fig 5J). However, this small enhancement of mCD8-mCherry expression in the anterior region was tiny compared to the enhancement of DptA following bacterial challenge, which was 14x higher in the anterior compared to the middle fat body on average (Fig 5C).

These results prove that the observed heterogeneity in DptA production during infection is not due to details of the microbial stimulus, but is a host-derived phenotype. Further, they demonstrate that the rate-limiting step of DptA production responsible for the cell-cell heterogeneity is downstream of peptidoglycan receptor activation. Supporting the latter finding, we saw no differences in levels of GFP-tagged PGRP-LC [49] between anterior and middle fat body regions prior to infection in wild-type larvae, though the weak expression levels in early L3 prevented accurate quantification (S10 Fig). The role of Imd in the heterogeneous antimicrobial peptide response is less clear cut. While the DptA expression patterns under ImdCA overexpression are more uniform than either of the PGRP-LCs, there is still a clear lack of response in the middle-dorsal lobes of the fat body, indicating that Imd activation is not the sole rate-limiting step for DptA production. Consistent with this notion, we observed uniform nuclear localization of a GFP-tagged Relish [52] across the fat body in wild-type larvae 6 hours post infection (S11 Fig), suggesting that the rate-limiting step of antimicrobial peptide induction is even downstream of nuclear import of Relish. However, unlike for PGRP-LC-GFP, we are unaware of rescue experiments showing that this tagged version of Relish functions equivalently to the endogenous protein, which limits the interpretability of our observations.

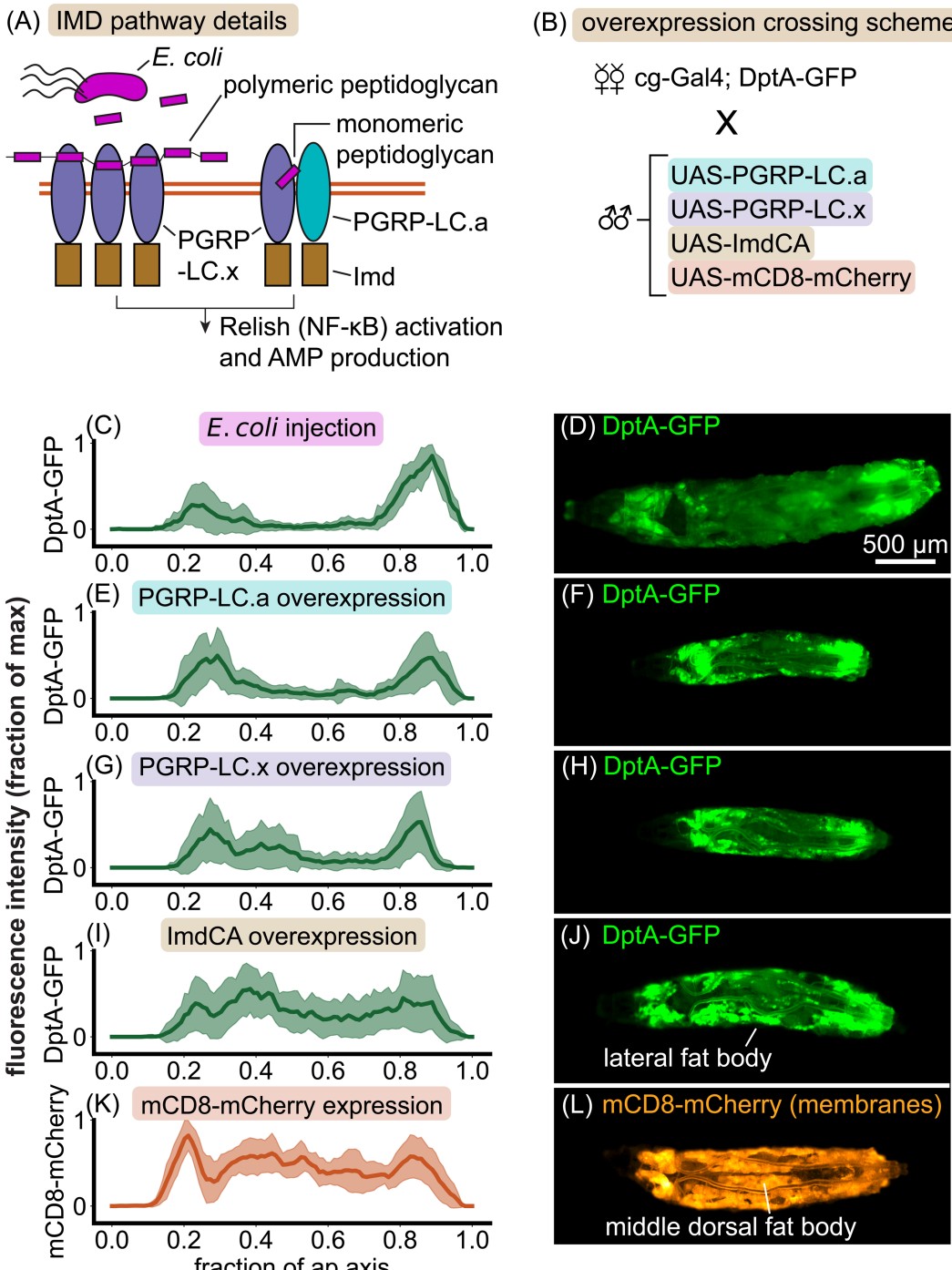

**Fig 5. Overexpression analysis reveals that the rate-limiting step of DptA production is downstream of peptidoglycan receptor activation.**
(A) Schematic of IMD pathway activation. Here we focus on two isoforms of a peptidoglycan recognition protein, PGRP-LC.x and PGRP-LC.a, which both in turn activate Imd. The cartoons are inspired by [47]. (B) Schematic of the cross used to express Imd signaling components in the fat body and read out the response of the DptA reporter. mCD8 is a membrane marker used as a positive control for uniform fat body activation. (C)-(L) Normalized fluorescence distributions along the anterior-posterior axis (left) and representative widefield images (right) of the various activation conditions. (C)-(D) *E. coli* injection generates a "U-shaped" expression profile (24 hours post injection; "partial responders" only, see Fig 1). $N = 10$ larvae. (E)-(F) PGRP-LC.a overexpression phenocopies the *E. coli* partial response. $N = 18$ larvae. (G)-(H) PGRP-LC.x overexpression produces a largely "U-shaped" expression

pattern, though with some additional expression in the lateral fat body. *N* = 15 larvae. (I)-(J) Overexpression of a constitutively active form of Imd, ImdCA [48], produces a largely uniform 1D expression profile along the anterior-posterior axis due to strong expression in the lateral fat body. However, larvae consistently lack expression in the middle dorsal lobes of the fat body. *N* = 22 larvae. (K)-(L) The membrane marker mCD8-mCherry shows a uniform expression profile, with small dips due to the anatomy of the fat body (see *Methods: Image analysis* for image analysis details). The dorsal-middle lobes of the fat body, which show the weakest DptA-GFP expression, are highlighted. *N* = 16 larvae.

Given that Ecdysone signaling leads to stronger DptA expression on average [22], we next asked if the observed spatial pattern in DptA expression could be explained by a spatial pattern of Ecdysone Receptor (EcR) nuclear localization. Ecdysone is secreted in its precursor form in pulses throughout the larval stage from the prothoracic gland [40], which is located in the anterior of the larva, near the anterior-dorsal lobes of the fat body that exhibit strong DptA expression. Therefore, we hypothesized that DptA expression in the anterior fat body might be explained by temporary spatial gradients in Ecdysone signaling. To test this hypothesis, we used a fly line containing an endogenously-tagged B1 subunit of Ecdysone Receptor, mNeonGreen-EcR-B1 (*Methods: Generation of mNeonGreen-EcRB1*). Levels of nuclear-localized mNeonGreen-EcR-B1 correlated with developmental stage, as expected (S12 Fig). Counter to our hypothesis, mNeonGreen-EcR-B1 concentration was largely uniform throughout the fat body (S13 Fig and S9 Movie), albeit with some local "patchiness" on the length scale of a few cells. Therefore, despite controlling the average DptA response across larvae over developmental time, these data suggest that EcR-B1 is not responsible for the observed variability in DptA expression within a single larva, though we have not ruled out the role of other EcR cofactors.

Altogether, these experiments show that the larval fat body contains distinct spatial microenvironments with heterogeneous immune activity, and that the mechanism limiting antimicrobial peptide production in the middle region of the tissue is downstream of peptidoglycan receptor activation, and possibly downstream of Relish activation. To explore the extent of this spatial compartmentalization of the larval fat body across other biological functions, we analyzed a previously published spatial transcriptomics dataset [38].

## Spatial transcriptomics reveals spatially patterned genes within the unperturbed fat body

To take a more unbiased approach to defining the spatial microenvironments of the fat body, we analyzed previously published, single-cell resolution spatial transcriptomics data of an unperturbed, early L3 larva obtained using StereoSeq [38]. In our quality checks (*Methods: Spatial transcriptomics analysis*), we found that the dataset accurately reproduced known spatial patterns of genes with posterior enrichment, including the Hox gene *abd-A* (S14 Fig) [31], indicating that the data accurately captures spatial patterning within the fat body. Sub-clustering fat body cells resulted in clusters that mapped to structurally and developmentally distinct tissue regions (Fig 6A–6B). In particular, the anterior-dorsal and posterior lobes of the fat body emerged as transcriptionally-distinct regions (Fig 6B, green and red regions, respectively). Differential expression analysis between these regions and the rest of the fat body resulted in hundreds of differentially expressed genes encompassing a wide range of biological processes (S15A–S15C Fig; S1 Data Data and S2 Data Data).

We cross-referenced our lists of differentially expressed genes with a list of 564 genes known to be involved in immunity [20]. For the anterior fat body, out of 721 upregulated genes, 31 were known immunity genes (Fig 6C). Strikingly, one of the top hits for defining this tissue region was Turandot-A (TotA; Fig 6D). TotA is a phosphatidylserine (PS) lipid-binding protein that protects host cells from antimicrobial-peptide-induced damage and apoptosis [53]. Other known immunity genes that define the anterior fat body include Nurf-38, a component of a nucleosome remodeling complex that regulates Ecdysone signaling [54], and lesswright, part of the sumoylation pathway that regulates several Imd and Toll pathway proteins [55] (Fig 6D). In the posterior fat body, 17/242 upregulated genes were known immunity genes (Fig 6E). Examples of posterior-enriched immunity genes include Materrazi (CG13905), which protects the host from infection-induced reactive oxygen species by binding lipids in the hemolymph [56], and two secreted peptides of unknown function, CG5773 and CG14957 (Fig 6F) [20].

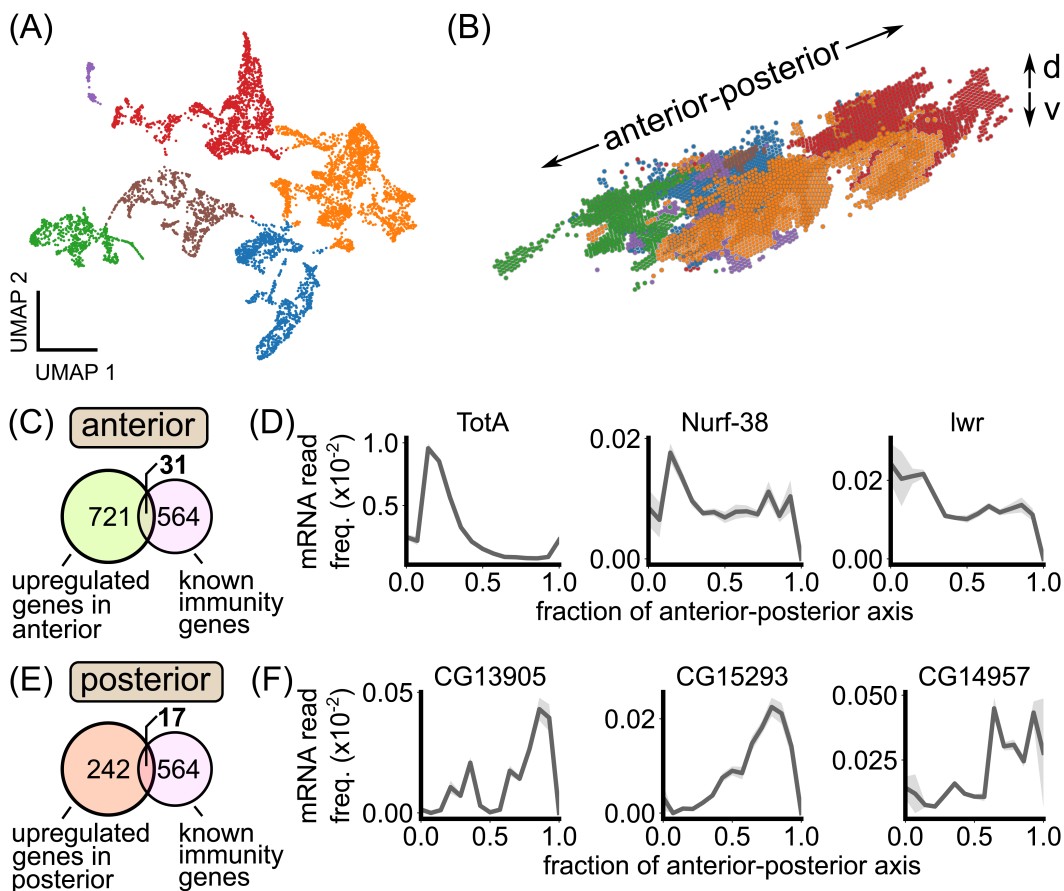

**Fig 6. Spatial transcriptomics reveals spatially patterned genes in the larval fat body, including known immunity factors.** (A) A two-dimensional representation ("UMAP") of fat body cell transcriptomes from the early L3 dataset from [38] colored by the output of a Leiden clustering algorithm. (B) 3D rendering of fat body cells in real space colored by the transcriptome clusters in (A). The transcriptome clusters correspond to distinct anatomical regions within the fat body. The anterior-posterior and dorsal-ventral ("d-v") axes are noted. (C) Overlap between genes upregulated in the anterior fat body and genes known to be involved in immunity. (D) Examples of immune-related genes that are upregulated in the anterior fat body. (E) Overlap between genes upregulated in the posterior fat body and genes known to be involved in immunity. (F) Examples of immune-related genes that are upregulated in the posterior fat body. In (D) and (F), expression profiles are plotted as the average transcript frequency across cells in each anterior-posterior axis bin, with error bars obtained from bootstrapping.

Together, these results support the existence of spatial immune microenvironments within larval fat body. In particular, the anterior and posterior regions of the fat body, which we identified to have enhanced antimicrobial peptide production, naturally emerged as transcriptionally distinct regions with marker genes including known immunity factors.

## Discussion

Using a new, quantitative live imaging approach, we characterized heterogeneous spatial patterns of immune activity the larval fly fat body [22–24]. Despite a strong correlation between the patterns of antimicrobial peptide expression and hemolymph flow, which could concentrate microbial stimuli, we found that the spatially heterogeneous response is independent of microbial localization. Rather, spatially heterogeneous antimicrobial peptide expression also occurred in over-expression experiments without infection, indicating that these patterns are intrinsic to the fat body tissue. More work is required to identify the precise mechanism limiting expression of antimicrobial peptides in the dorsal-middle lobes of the

fat body and why it primarily affects the Imd but not Toll pathway. As a first step, we have determined that the mechanism acts downstream of peptidoglycan receptor activation. Imaging of fluorecently tagged transcription factors indicated that two major proteins known to play key roles in the transcription of Imd-dependent immunity genes, Relish and the B1 subunit of Ecdysone Receptor, are not the limiting factors. Investigating additional Ecdysone Receptor cofactors [57,58] would be a logical next step, as would testing the role of Hox genes [59]—master regulators of anterior-posterior patterning in the embryo—in regulating immunity genes in the larval stage.

While the correlation between immune activity and blood flow was not causal, we speculate that these regions are primed for immune response because they sit in regions of high microbial exposure via blood flow, consistent with the notion of "functional integration" between circulatory and immune systems [60], and analogous to the concentration of leukocytes at the portal vein of the mammalian liver [61]. This spatial configuration also resembles the structure of lymph nodes, where sentinel macrophages line the lymph node interface and upon infection rapidly relay signals to adjacent lymphocytes [1]. Our finding that, within the spatial transcriptomics data from [38], the host-protective factor TotA expression is strongly biased to the anterior fat body, where antimicrobial peptide expression is generally the strongest, suggests an intriguing co-regulation mechanism for minimizing self-damage during immune response. This situation conceptually resembles the landscape of the intestine, where sentinel dendritic cells monitor microbial activities in the gut and can induce both proinflammatory and tolerogenic responses [62]. In addition to mirroring patterns of blood flow, proximity to key organs (especially for the anterior lobes, which sit near the central nervous system, imaginal disks, and other important structures), may also explain the logic behind spatial compartmentalization of immune activity. More generally, spatial compartmentalization of functions within the fat body may be advantageous for this highly multifunctional tissue, which, in addition to its role in immunity, performs essential metabolic and enzymatic tasks [33,34].

One limitation of our results is that our observations are confined to the early third instar stage. While this developmental stage is short lived compared to the life of the fly (~1 day compared to ~1 month), it is also one that has a strong susceptibility to infection, given the immersion in fermenting substrates and predation from parasitoid wasps that may result in microbial co-infection [63]. The extent to which the spatial patterning of antimicrobial peptide expression occurs in adult flies has only begun to be explored. Recent work using single-nucleus RNA sequencing revealed multiple subtypes of adult fat body cells with distinct immunological characteristics, though their spatial configuration remains to be determined [32]. Expression of antimicrobial peptides following *Providencia rettgeri* infection was largely uniform across adult fat body cells [32], which suggests that the spatial patterning of antimicrobial peptide expression may be restricted to early larval stages. However, further testing across microbial stimuli and doses is required for a broader characterization of possible expression patterns in adult flies.

Finally, we emphasize that the live imaging approach introduced here constitutes a significant improvement in the ability to quantify gene expression dynamics during immune responses with a large field of view and single-cell resolution, for any organism. Previous pioneering examples in flies [64] and zebrafish [65] established in vivo, single-cell imaging of fluorescent reporters of gene expression during infection and immune cell differentiation, but were limited to only a few cells at a time. With light sheet fluorescence microscopy, we are able to image over 1000 cells for several hours at 2 minute intervals, significantly expanding the possibility of studying organism-scale immune response dynamics at single-cell resolution. Advances in light sheet microscope design that simplify sample mounting [66,67] will no doubt improve the feasibility and throughput of such measurements.

## Methods

### Fly stocks

Antimicrobial peptide reporter lines were from [21]. Specifically, we obtained DptA-GFP as a kind gift from Neal Silverman; Drosomycin-GFP from Bloomington (BSDC 55707); Attacin-GFP, Cecropin-A1-GFP, Defensin-GFP, Drosocin-GFP, and Metchnikowin-GFP were kind gifts from David Bilder and Stephan Gerlach. Fat body Gal4 drivers were r4-Gal4 and

cg-Gal4, kind gifts from David Bilder. Membranes were marked with UAS-mCD8-mCherry (BDSC 27391). Histones were marked with UAS-His-RFP, a kind gift from Jack Bateman. The larval heart Gal4 driver was NP1029-Gal4, a kind gift from Rolf Bodmer and Erick Eguia. Heartbeat knockdowns were done with UAS-Mhc-RNAi (BDSC 26299) and UAS-Ork1DeltaC (BDSC 8928). Peptidoglycan receptor overexpression lines UAS-PGRP-LC.a (BDSC 30917) and UAS-PGRP-LC.x (BDSC 30919) were obtained from Bloomington. Constitutively active Imd, ImdCA, was from [48]. The GFP-tagged PGRP line was from [49] and was a kind gift from Bruno Lemaitre. The GFP-tagged Relish line (BDSC 81268) was from [52] and was obtained from Bloomington. Hemocytes were marked with srpHemo-3XmCherry (BDSC 78358).

## Generation of mNeonGreen-EcR-B1

The endogenous mNeonGreen-EcR-B1 fusion line was generated by CRISPR-Cas9 genome editing. A DNA mixture was prepared containing 500 ng/μl of mNeonGreen-EcR-B1 homology donor (pTG614), 500 ng/μl of Halo-EcR-B1 homology donor (pTG609; not used in the present study), 300 ng/μl of EcR-B1 U6-sgRNA plasmid (pTG613), and 200 ng/μl of ebony control U6-sgRNA plasmid (pTG625). This mixture was injected by Rainbow Transgenic Flies, Inc. (Camarillo, CA) into a fly line containing a germline-expressed nos-Cas9 transgene at the attP2 locus (chromosome 3). Injectants were crossed to the Sp/CyO; Dr/TM3,ebony(-) double-balancer line, and progeny from vials containing ebony(-)/TM3,ebony(-) flies were crossed to Sp/CyO to establish balanced lines. Successful insertion of mNeonGreen was confirmed by PCR and Sanger sequencing, and the knock-in was made homozygous. The ebony control U6-sgRNA plasmid was a kind gift of Colleen Hannon, and the empty U6-sgRNA plasmid was a kind gift of Mike Stadler.

## Bacteria and Yeast

*E. coli* HS-tdTomato [68] was used for all experiments and was a gift from Travis Wiles and Elena Wall. For every experiment, bacteria were grown fresh overnight shaking at 37°C. *S. cerevisiae* strain SK1 (non-flocculating mutant) with marker HTB1-mCherry-HISMX6 was a gift from Tina Sing. Yeast were streaked on YPD plates and grown overnight at 30°C then picked and grown in YPD liquid culture overnight shaking at 30°C. The full genotype of the yeast strain was MATa, ho::LYS2, lys2, ura3, leu2::hisG, his3::hisG, trp1::hisG, flo8 unmarked, amn1(BY4741 allele)unmarked, HTB1-mCherry-HISMX6, GAL3+. *P. rettgeri* strain Dmel and *E. faecalis* were kind gifts from Brian Lazzaro. Both strains were grown at 37°C in LB.

## Larva collection

Flies were placed in a fresh food vial for 24 hours and then kept for 4 days at 25°C. Larvae were collected via flotation using 20% sucrose solution for no more than 5 minutes. Unless otherwise specified, late L2 larvae were identified by anterior spiracle morphology (containing hybrid L2/L3 spiracles) and placed in a fresh food vial for 6 hours at 25°C. Molt to third instar was confirmed after 6 hours, after which larvae were placed in another fresh food vial. Larvae were then stored according to their age treatment. Most experiments had larvae placed in 18°C for 18 hours ("18h-18°C"). In all experiments, larvae were handled gently with a fine paintbrush to avoid potential immune response activation via mechanical stimulation [69].

## Larva anesthesia

For injections and imaging, larvae were subjected to ether anesthesia as described by [70]. In brief, an anesthesia chamber was constructed out of a Coplin staining jar filled with cotton balls and a small glass vial. The cotton was supersaturated with diethyl ether inside of a chemical fume hood. A small cage was made out of a cut top of an Eppendorf tube and fine mesh. Larvae were placed in the cage and the cage was placed in the small glass vial within the anesthesia chamber for a prescribed amount of time. For injections, a batch of around 10 larvae were anesthetized for 2 minutes and 15 seconds. For time-lapse imaging, individual larvae were anesthetized for 45 seconds prior to mounting in glue (see below).

For endpoint imaging, larvae were fully immobilized using 3 minutes and 45 seconds of anesthesia exposure. We note that in our experience, the effect of the ether anesthetic on larvae could be quite variable, being sensitive especially to larval humidity and density, and so in some cases was adjusted to obtain the desired effect.

## Microinjection

To prep the injection mix, 1 ml of overnight bacteria or yeast culture was centrifuged for 2 minutes in a small centrifuge at 8000 RPM, washed once, and resuspended in 200 $\mu$l of 0.2% sterile saline solution. The injection mix contained a 1:1 mix of this bacterial or yeast solution with a 1 mg/ml solution of Cascade Blue Dextran, which acts as a fluorescent marker of injection success.

Microinjections were performed using a Narigishe IM 300 microinjector under an Olympus SZX10 fluorescent stereo-microscope. Fine-tipped quartz glass needles were pulled on a Sutter P-2000 pipette puller using 0.7 mm ID/1.0 mm OD quartz glass needles with filament (Sutter item num. QF100-70-7.5). Pulled needles were filled with injection mix using a micropipette loader tip and then inserted into a needle holder mounted on a 3-axis micromaniuplator. The needle was gently broken on the edge of a glass slide, producing a 5-10 $\mu$m sized tip (S1 Fig). We found that quartz glass was required to obtain a needle that was both fine and rigid enough to easily penetrate the larval body wall. Injection droplet size was calibrated to a 300 $\mu$m diameter using a sterilized stage micrometer and was periodically checked throughout an injection session.

Batches of around 10 larvae were anesthetized for 2 minutes and 15 seconds and then mounted on a sterilized glass slide dorsal side up. Prior to injection, larva health status was assessed by looking for a normal heartbeat and minor mouth hook movements (for experiments involving loss of heartbeat, just minor mouth hook movements were used as a marker of health). Except for the injection location control experiments (S2C–S2F Fig), injections were done on right side of the body wall between segments 5 and 7, avoiding the fat body itself. Needle penetration was done under a low-intensity brightfield light, but then the light sources was switched to a blue fluorescence channel for the actual injection. The needle was held in place for 10 seconds and the blue dye was observed to confirm a normal flow pattern: the dye as a bulk shifts to the posterior and then dye can be seen being pumped through the larval heart. Animals with abnormal flow patterns were discarded, as were any animals for which significant dye leaked out of the injection site after needle removal. After successful injection, larvae were placed in a humid Petri dish. Using this injection method with fine-tipped quartz needles, we observed no melanization response common to other infected wound models.

## Light sheet fluorescence microscopy

Three-dimensional images were acquired using a Zeiss Z.1 Light Sheet Fluorescence Microscope. Two different configurations were used in this paper: (1) 20x/1.0 NA water dipping detection objective with 10x/0.2 NA illumination objectives and (2) 5x/0.16 NA air objective with 5x/0.1 NA illumination objectives. The detection objective used for each experiment is listed below. For all experiments and for each z-plane, images were acquired with both excitation sheets in rapid succession and then later averaged. All experiments used pivot scanning to reduce striping artifacts.

**Single time point imaging with the 20x water objective.** The 20x water configuration was used for single-time point images only. Larvae were immobilized with ether and embedded in a 1% agarose gel pulled into a glass capillary. Laser power was 5% maximum for both 488nm and 561nm channels. Exposure time was 30 ms, light sheet thickness was set to 6.5 $\mu$m, and z-slices were acquired every 2 $\mu$m. To capture the full larva width, a zoom of 0.7 was used and the light sheet thickness was extended to 6.5 $\mu$m. However, at this zoom, the light sheet incompletely filled the detection plane in the vertical direction, leading to low-intensity artifacts at the top and and bottom of images. Therefore, images were cropped in vertical direction. In addition, remaining low-intensity artifacts were corrected by normalizing images by a fit to a reference image obtained by average several pictures of uniform fluorescence (for green fluorescence, a solution of pure EGFP and for red fluorescence, a solution of rhodamine) in agarose. We fit an intensity field of

$$I(x, y) = \frac{I_0}{1 + \left(\frac{x-x_c}{x_R}\right)^2} e^{-\frac{(y-y_c)^2}{2\sigma_y^2}} \tag{1}$$

where $x$ is the sheet propagation direction and $y$ is the vertical direction.

Images taken with the 20x water objective: Figs 1C, Fig 4H, Fig 4J, S12, D and S13.

**Time lapse imaging with the 5x air objective.** As fly larvae cannot receive sufficient oxygen while submersed in water, imaging of these samples for longer than a few minutes on classical light sheet microscopes, which rely on immersion in a refractive medium, poses a technical challenge. Our solution was to use halocarbon oil as an immersive medium. Halocarbon oil is rich in oxygen and larvae can survive for over 24 hours fully submerged in it, albeit in a reduced oxygen environment. We filled the sealed imaging chamber with halocarbon oil 27 and the 5x air objective was placed outside of glass window of the chamber. Halocarbon oil 27 has a refractive index of 1.4. To align the Zeiss light sheet in this non-conventional imaging media, we used the objective adapter designed for $n = 1.45$ clearing media together with light sheet galvo mirror settings designed for water immersion.

To mount larvae for timelapse imaging, larvae were anesthetized with ether for 45 seconds and then glued ventral side down onto 2 mm acrylic rods, which were mounted into the standard Zeiss light sheet sample holder. The glue used was Elmer's washable clear glue, as was done in a previous protocol for adult fly imaging [71]. The glue was applied in three layers. First, a thin layer was used as base to secure the larva and let dry for 3-5 minutes. Then, a layer was applied to each side of the larva, making contact between the lateral body wall and the acrylic rod, and let try for 3-5 minutes. Finally, a layer was applied on the dorsal side of the larva, bridging the two lateral glue layers and avoiding the posterior spiracles, letting dry for 3-5 minutes. This gluing method constrained larval movement and produced minimal aberration on the low-NA 5x objective.

Laser power was 30% of maximum for both 488nm and 561nm channels. Exposure time was 30 ms, light sheet thickness was set to 8.16 $\mu$m, and $z$-slices were acquired every 4 $\mu$m.

Images taken with the 5x air objective: Figs 2C–2D, 3A, Fig 4B and A–C.

## Widefield microscopy

Low magnification, widefield images of antimicrobial peptide expression patterns were obtained on a Zeiss AxioZoom fluorescence microscope. Larvae were immobilized with ether, mounted on a glass slide dorsal side up, and imaged using a 1X objective using a zoom of 29.5X, an exposure time of 10ms, and an LED power of 100% on an XCite light source.

## Image analysis

**Image registration and *zarr* conversion.** Images for each time point, tile, and light sheet illumination were saved as separate .czi files and then assembled using custom Python code. Images from the two sheet illuminations were combined with a simple average. For images taken with the 20x objective, each fused $z$-plane was corrected for sheet intensity (see "Light sheet fluorescence microscopy" section above). Images from different tiles were registered using stage coordinates extracted from the .czi file using the `aicsimageio` package [72]. The final image was saved as a 5-dimensional OME-Zarr file [73].

**Single-cell DptA-GFP expression levels from a membrane marker.** Single-cell DptA-GFP levels from Fig 1 were quantified in 2D maximum intensity projections. In our initial experiments, we aimed to segment fat body cells based on a fluorescent membrane marker, r4>mCD8-mCherry. However, we found that mCD8-mCherry was additionally localized to the periphery of lipid droplets within fat body, which complicated membrane segmentation. Therefore, we took a manual approach and used the interactive visualizaton program `napari` [74] to click on cell centers. Before maximum intensity

projection, the membrane signal was enhanced using a UNet model from PlantSeg ("2dunet_bce_dice_dx3x") [75]. GFP Fluorescence intensity was summed within a circle of radius of 6 pixels ($\approx 2\mu$m) around the manually-defined cell center.

**Spatial patterns of DptA-GFP expression along the anterior-posterior axis.** To quantify tissue-scale spatial patterns of DptA-GFP expression in the absence of a fat body cell marker, we used Multi-Otsu thresholding of 2D maximum intensity projections. Specifically, we computed 2 Otsu thresholds of log-transformed intensity images, resulting in three image categories with typically well-spaced log-intensity peaks: dim background, bright background, and strong GFP signal. We then thresholded on the strong GFP signal and summed along the short axis of the larva to obtain a 1D intensity distribution along the anterior-posterior axis (Fig 1D). For quantification of spatial patterns from widefield microscopy images, thresholds were chosen manually. The same threshold was used for each reporter.

**Quantification of DptA-GFP expression dynamics.** In our timeseries imaging experiments, we used a nuclear marker, r4>HisRFP. Due to rapid motion from larval twitching and internal hemolymph flows, nuclei were tracked manually in 2D maximum intensity projections using `napari` [74]. GFP Fluorescence intensity was summed within a circle of radius 6 pixels ($\approx 5.5\mu$m) microns around the manually-defined cell center.

**Quantification of nuclear-localized Ecdysone receptor levels.** Fat body nuclei (r4>HisRFP) were segmented in 3D using straightforward thresholding after Gaussian blur. Parameters were tuned such that *E. coli*-tdTomato, though visible in the images, were not segmented due to being much smaller and dimmer than fat body nuclei. Segmentation was done in Python using the GPU-powered package `cucim` followed by CPU-based labeling using `scikit-image`. Ecdysone receptor levels (mNeonGreen-EcR-B1) were then quantified by subtracting local background fluorescence around each nuclei, obtained by averaging the pixel values in a shell around each nucleus of obtained by dilating the nuclear mask by 2 pixels and subtracting the original mask, then summing the green channel fluorescence intensity within each nucleus.

**Bacteria segmentation.** Bacteria were segmented in two phases, similar to the approach of [76]. First, single-cell and small bacterial clusters were identified by Difference of Gaussians filtering and thresholding. Then, larger bacterial clusters, which here often appear to be inside of nephrocytes and hemocytes, are segmented by Gaussian blurring and thresholding. The two masks are computed on the GPU, then combined and resulting mask is used to compute a label matrix on the CPU. We then compute the summed fluorescence intensity of each object in the label matrix and estimate number of bacteria per object by normalizing by the median intensity and rounding up to the nearest integer. We chose the median intensity as a normalization factor based on visual inspection of the images and corresponding fluorescence intensities of each object. We defined planktonic bacteria as clusters with a size less than 3 cells, which we determined by visual inspection to most accurately capture single-cells.

**Computer specifications.** Image analysis was done on a custom-built workstation with an Intel Core i9 11900K processor, GeForce RTX 3070 8GB GPU, and 128 GB RAM running Ubuntu 20.04.

### Heartbeat knockdown experiments

We used two strategies to eliminate the heartbeat. First, following the work of reference [45], we over-expressed the potassium channel Ork1 using the larval heart-specific driver NP1029. This scheme produced robust elimination of the heartbeat (S8 Movie, left). However, we found that starting approximately 3 hours after either bacteria or mock injections, the heart began beating again and by 6 hours was steadily beating in the majority of larvae (S8 Movie, right). The mechanism behind this effect is unknown. As this timescale of regaining a heartbeat interfered with our immune response measurements, we turned to a more severe perturbation. Following reference [46], we knocked down myosin heavy chain in the larval heart via NP1029>Mhc-RNAi. We found that this scheme eliminated the heartbeat in a manner robust to injection (S4 Movie).

For the characterization of bacterial spatial distribution and fluid flows in heartbeat-less animals (Fig 4), animals were reared at 25°C and staging was less precise—we simply picked early third instar larvae out of the food.

For the measurement of Diptericin expression in heartbeat-less animals and matched controls, we used a trans-heterozygote scheme described in S8 Fig. F2 larvae were screened for or against the presence of a heartbeat under a dissection microscope. Animals lacking any detectable GFP expression after infection were discarded. In this experiment, larvae were staged precisely according to the 18 hours post L3 molt at 18°C protocol described above. To maximize the effect of the RNAi while including this period at 18°C, larvae were raised from egg laying to late L2 at 29°C (see schematic in S8B Fig). Heartbeats (and lack thereof) were monitored throughout the experiment: before and after molt to L3, before and after injections, and before mounting for imaging. Larvae that were first identified as having no heartbeat but later exhibit some beating were discarded. Larvae that were identified to have a heartbeat but lacked a heartbeat after ether exposure prior to injections were also discarded.

### Imd pathway overexpression experiments

We crossed *cg*-Gal4; DptA-GFP females to males carrying UAS-driven components of the Imd pathway, including PGRP-LC.a, PGRP-LC.x, and ImdCA. The crosses were performed at 25°C until late L2, where larvae were screen and placed at 18°C overnight, following our main staging protocol for early L3. Larvae were imaged 3-6 hours after they would have been injected in our infection experiments. A membrane marker, UAS-mCD8-mCherry, was used as a positive control for a uniform expression pattern.

### Spatial transcriptomics analysis

StereoSeq data, in the form of a processed anndata file, of an unpertrubed, early L3 larva was obtained from [38]. Analysis was done using the scanpy package [77]. Fat body cell annotations were taken directly from [38]. Reads were further filtered to 5% detection. Leiden clustering was performed on analytic Pearson residuals [78] with parameters: $n_{iterations} = 2$, $resolution = 0.04$. The resolution was chosen by starting with a low value and increasing until the anterior fat body emerged as a cluster. Differential expression analysis was done on the preprocessed matrix from [38], output from SCTransform. Marker genes for each cluster were found using the Wilcoxon test using Bonferonni correction and are included in S1 Data and S2 Data Data. For plotting one-dimensional expression patterns along the anterior-posterior axis, we computed the average transcript frequency within each anterior-posterior axis bin. Error bars were obtained by bootstrapping this average over 100 groups of cells sampled with replacement within each bin.

## Supporting information

### Supplemental Movies

**S1 Movie 3D rendering of a "partial response" larva expressing DptA-GFP (green) 24 hours after injection with *E. coli*-tdTomato.** The DptA-GFP channel has been log-transformed for visual clarity. Also shown in magenta is a fat body membrane marker, r4>mCD8-mCherry. Anterior is to the left. Scale bar is 500 $\mu$m. See also Fig 1C,i.
(MP4)

**S2 Movie Timeseries of maximum intensity projections showing the initial activation of DptA-GFP (green).** Fat body nuclei are marked in magenta via cg>His-RFP. Movie starts 5 hours post injection with *E. coli*-tdTomato. Anterior is to the left. Scale bar is 500 $\mu$m. See also Fig 3A.
(MP4)

**S3 Movie Timeseries of maximum intensity projections showing the initial activation of DptA-GFP (green).** Fat body nuclei are marked in magenta via cg>His-RFP. Movie starts 6 hours post injection with *E. coli*-tdTomato. Anterior is to the left. Scale bar is 500 $\mu$m.
(MP4)

**S4 Movie Real time movies of heartbeats visualized by green autofluorescence in a wild-type larva (left) and in a larva in which myosin was knocked down in the heart (NP1029>Mhc-RNAi).** Heart-specific myosin knockdown eliminates the heartbeat but still allows larva motility and body contractions. Anterior is to the left. Scale bar is 250 $\mu$m.
(MP4)

**S5 Movie Real time movie of *E. coli*-tdTomato transport in blood flow.** Bacteria can be seen being pumped directly through the heart from posterior to anterior (right to left) and then returning via retrograde flow outside the heart (left to right). Anterior is to the left. Scale bar is 250 $\mu$m.
(MP4)

**S6 Movie 3D renderings of *E. coli*-tdTomato (top, magenta) 3 hours post-injection and the corresponding computational segmentation (bottom, colors).** Anterior is to the left. Scale bar is 500 $\mu$m.
(MP4)

**S7 Movie Real time movie of *E. coli*-tdTomato transport in the hemolymph of a larva lacking a heartbeat.** Bacteria can be seen being pushed around by body wall contractions, but are not pumped through the heart. Loss of heartbeat was achieved by knocking down myosin heavy chain (Mhc) in the heart via the genotype NP1029-Gal4; UAS-Mhc-RNAi; +. Anterior is to the left. Scale bar is 250 $\mu$m.
(MP4)

**S8 Movie Real time movies of hearts visualized by green autofluorescence larvae in which heartbeats were disrupted by heart-specific overexpression of the potassium channel, Ork1 (NP1029>Ork1) [45].** Despite successful elimination of the heartbeat via Ork1 overexpression (left), mock microinjection (right) restarts the heart by 6 hours post-injection. Anterior is to the left. Scale bar is 250 $\mu$m.
(MP4)

**S9 Movie 3D rendering of mNeonGreen-EcR-B1 levels (cyan) in fat body nuclei (magenta, cg>His-RFP) 18 hours post molt to L3 at 18°C.** Anterior is to the left. Scale bar is 500 $\mu$m.
(MP4)

**Supplemental Data Files**

**S1 Data CSV file of genes that are differentially expressed in the anterior fat body (Leiden cluster 2).**
(CSV)

**S2 Data CSV file of genes that are differentially expressed in the posterior fat body (Leiden cluster 3).**
(CSV)

**S3 Data ZIP of imaging-based data files.**
(ZIP)

**Supplemental Figures**

**S1 Fig Details of microinjections.** (A) Brightfield image of an injection needle showing the taper. (B) Brightfield image of the needle tip, which ranges from 5-10 $\mu$m. (C) Fraction of larvae showing detectable DptA-GFP expression on a

low-magnification widefield microscope as a function of injection dose, in terms of dilution factor of the initial inoculum. The inoculum contains on average $10^5$ E. coli cells. Number of larvae per group: 7, 18, 12, and 13, for 1000x, 100x, 10x, and 1x dilution factors, respectively.
(EPS)

**S2 Fig Expression of DptA-GFP after microinjection of E. coli produces a repeatable, quantitative spatial pattern that is independent of injection site.** (A) Quantification of total DptA-GFP fluorescence intensity 24 hours post infection from a widefield microscope in animal heterozygous and homozygous for the reporter. While there is strong animal-animal variability, the median intensity of homozygotes ($7.4 \cdot 10^4$ a.u.) is close to twice the median intensity of heterozygotes ($3.2 \cdot 10^4$ a.u.), as expected. (B) Total DptA-GFP fluorescence intensity of larvae in which either ether or cold shock ("ice") was used for immobilization during injection. The two immobilization methods produce distributions of total DptA-GFP expression that are comparable within error. (C)-(F) The observed spatial pattern of DptA-GFP expression is independent of injection site. Larvae were injected at 3 different locations, "anterior left", "middle right", and "posterior right" (noted by white arrow heads in the images) and were assessed for DptA-GFP expression 24 hours later on a widefield microscope (single z-plane widefield images shown in (C)-(E), quantification in panel (F), mean and standard deviation of fluorescence intensity normalized to the maximum value for each larva across anterior-posterior bins. N =7 larvae for anterior, 5 for middle, 6 for posterior). The contrasts of images in (C)-(E) were adjusted identically; the apparent saturation does not reflect saturation of the camera, but was chosen to highlight regions of weak expression.
(EPS)

**S3 Fig Additional details of the partial DptA response.** (A) Partial-complete split by different metadata. From left to right: original clustering by single-cell median DptA-GFP fluorescence intensity produces a clean separation; larva sex, where we see a partial correlation in that all the males observed are partial responders; length of the fat body, which is a proxy for developmental stage and thus Ecdysone levels, though we see no correlation; experiment date, to control for effects related to the details of experiment preparation and injections, where we see no correlation. (B) Normalized DptA-GFP distributions along the anterior posterior axis shown for each larva in the partial responses group. (C) Probability densities of single-cell DptA-GFP fluorescence intensity for each larva in the partial responses group.
(EPS)

**S4 Fig A spatially heterogeneous AMP response occurs during microinjection-induced infection with P. rettgeri but not with E. faecalis.** Left panels: fluorescence intensity profiles of AMP reporters along the anterior-posterior axis, normalized to the maximum within each larva and then averaged. Shaded error bars represent standard deviations across larvae. Right panels: widefield microscopy images of two larvae per infection experiment, chosen to represent the diversity of expression patterns observed. (A)-(C) P. rettgeri induces a "U-shaped" expression pattern of DptA-GFP on average, mirroring the response to E. coli (main text Fig 1). Some animals exhibit only expression in the anterior fat body (C). (D)-(F) E. faecalis, a gram-positive bacteria containing Lys-type peptidoglycan that activates the Toll pathway, induces the Toll-responsive AMP Drosomycin (Drs) in an expression that is homogeneous across the fat body. However, some animals showed weak fat body expression, with the GFP signal being dominated by expression in the tracheal system; all animals were pooled for quantification, which accounts for the slightly different shape of the profile in (D) compared to (G). (G)-(I) E. faecalis also induces uniform expression profiles in the Toll-responsive AMP Metchnikowin (Mtk), with less variability across larvae compared to Drs-GFP. All imaging was done 6 hours post injection of bacteria. Approximately 10% of larvae died by this time point within each experiment and were excluded from the analysis.
(EPS)

**S5 Fig Dynamics of DptA activation on a conventional widefield microscope mirror findings using light sheet fluorescence microscopy.** (A) Single *z*-plane images of the anterior of a larva carrying the DptA-GFP reporter taken at 2.5 hours post injection (left) and 5 hours post injection (right). (B) Quantification of DptA-GFP fluorescence intensity in the anterior fat body over time (mean and standard deviation over $N = 11$ larvae).
(EPS)

**S6 Fig Gallery of bacterial localization.** (A) Single *z*-plane image planktonic bacteria (*E. coli*, marked by tdTomato) in the hemolymph (arrow heads). These cells were identified as suspended freely in the hemolymph by their motion in subsequent *z*-planes. (B) Single *z*-plane image showing *E. coli* on the posterior end of the heart. (C) Maximum intensity projection image showing an example of *E. coli* co-localizing with known patterns of sessile hemocyte bands (arrow heads) [64]. (D) Maximum intensity projection image showing an example of *E. coli* internalized by nephrocytes embeded in the heart. Images in panels (A)-(C) are from 3-5 hours post injection. Panel (D) is from 6-8 hours post injection.
(EPS)

**S7 Fig Additional details of bacteria spatial distribution quantification**. (A) Spatial distribution of all bacteria. (B) Input-output functions for DptA-GFP vs. planktonic bacterial density for the anterior (orange) and posterior (magenta) fat body. The two regions were defined from the peak of DptA-GFP expression to the middle of the fat body, with one anterior-posterior axis bin 0.07x the length of the fat body, or around 160 $\mu$m) separating the regions.
(EPS)

**S8 Fig Details of the heartbeat knockdown experiment.** (A) Fly crossing scheme for generating flies lacking a heartbeat and containing the DptA reporter. (B) Schematic of the timeline and temperatures used in the heartbeat knockdown experiment.
(EPS)

**S9 Fig Temporary loss of heartbeat by overexpression of Ork1 results in no change in the spatial pattern of DptA-GFP expression.** (A)-(B) Single z-plane widefield images of larvae 24 hours after infection for larvae that either had (A) or did not have (B) a heartbeat at the time of injection. Loss of heartbeat was achieved via the larval heart-specific driver NP1029>Ork1 (the crossing scheme was identical to the scheme in S8A Fig). Starting approximately 3 hours after injection, heartbeats begin to beat again (S8 Movie). (C) Quantification of DptA-GFP spatial pattern (mean and standard deviation across $N = 6$ larvae for heartbeat$^+$ group, $N = 8$ for heartbeat$^-$ group) normalized to the maximum value for each larva across anterior-posterior bins.
(EPS)

**S10 Fig Mapping the expression of PGRP-LC-GFP suggests weak but uniform expression of the peptidoglycan receptor across the fat body.** (A) Schematic of immune-relevant tissues in a fly larva. (B)-(F) Maximum intensity projections of PGRP-LC-GFP images taken using light sheet fluorescence microscopy. The fluorescently tagged receptor is the BAC line from [49]. Since expression levels vary widely across tissues, image contrast is adjusted separately for each image for visual clarity (except (C) and (D), which are identical). Quantification of signal intensity is given in (G). (B) Middle fat body of a wandering L3 larva, which shows high levels of expression and protein clusters. (C) Anterior fat body of an early L3 larva (18 hours post molt to third instar at 18°), showing weak expression and fewer detectable clusters. (D) Middle fat body of the same larva in (C), showing similar expression levels as the anterior fat body. (E) We also observe weak PGRP-LC-GFP expression in hemocytes (marked by srpHemo-3xmCherry), which are especially prominent in the posterior sessile clusters [64]. (F) We observe strong expression of PGRP-LC-GFP in the lymph glands. (G) Quantification of fluorescent signal intensity in each of the images in (B)-(F). Signal was extracted via multi-Otsu thresholding with

3 categories—representing image background, tissue background, and tissue signal—and summing pixel intensity in the high intensity category.
(EPS)

**S11 Fig A transgenic GFP-Relish line shows uniform nuclear localization patterns across the fat body.** (A) GFP-Relish shown as maximum intensity projection from light sheet fluorescence microscopy images 6 hours post infection with *E. coli*. The GFP-Rel construct is in a BAC from [52]. In addition to the fat body, we see activation of GFP-Rel in hemocytes of the lymph gland and posterior sessile clusters. (B) Quantification of GFP-Rel fluorescence intensity (fraction of maximum) along the anterior-posterior axis. Shaded lines indicate standard deviation across $N = 4$ larvae. We used Difference of Gaussians filtering and thresholding to segment fat body nuclei apart from the hemocytes. The profile is uniform along the length of the fat body.
(EPS)

**S12 Fig Nuclear mNeonGreen-EcR-B1 fluorescence intensity correlates with developmental stage** (A)-(B) Maximum intensity projections of mNeonGreen-EcR-B1 images in early (A) and late (B) third instar larvae. Image regions correspond to areas approximately above the anterior-dorsal lobes of the fat body, but the images are not masked by fat body nuclei, so they contain signal from multiple cell types.
(EPS)

**S13 Fig Nuclear localization of Ecdysone receptor (EcR), a transcriptional regulator of Diptericin, does not correlate spatially with DptA-GFP expression.** (A) An example single z-slice of middle-dorsal fat body nuclei showing raw fluorescence of mNG-EcR-B1 (left, cyan), fat body histones marked by cg-Gal4; UAS-HisRFP (middle, magenta), and the merged image. (B) Maximum intensity projection of a full view of the larval fat body showing mNG-EcR-B1 localization. While the localization pattern exhibits some degree of local structure, unlike DptA-GFP, it is broadly uniform along the anterior-posterior axis. Fat body nuclei were computationally segmented and then false colored (cyan channel) by their mean background-subtracted mNG-EcR-B1 fluorescence intensity (*Methods: Quantification of nuclear-localized Ecdysone receptor levels*). In this way, EcR levels in non-fat body cells are not visualized. Note that cg-Gal4 also labels hemocytes, but hemocytes are computationally removed based on their smaller size (*Methods: Quantification of nuclear-localized Ecdysone receptor levels*). (C) Quantification of nuclear-localized mNG-EcR-B1 levels along the anterior-posterior axis. Solid line and shaded error bars are the mean and standard deviation respectively across $N = 6$ larvae.
(EPS)

**S14 Fig Spatial transcriptomics data recapitulates known genes with posterior peaks**. These genes were identified in a differential expression analysis of bulk RNA-seq from dissected tissue fragments and being enriched in the posterior compared to the middle-lateral fat body [31]. The first gene, *abd-A*, is a Hox gene involved in anterior-posterior patterning. Some genes also exhibit a peak in the anterior fat body, which was not included in [31].
(EPS)

**S15 Fig Additional details of spatial transcriptomics analysis.** (A) 3D rendering of fat body cells colored and labelled by Leiden cluster. (B) Matrix plot showing the top 5 marker genes defining each cluster.
(EPS)

## Acknowledgments

We thank Megan Martik, Stephan Gerlach, Yoshiki Sakai, David Bilder, Rolf Bodmer, Erick Eguia, Travis Wiles, Elena Wall, Noah Whiteman, Colleen Hannon, and Mike Stadler for access to equipment and/or reagents. We thank Dennis Sun for advice on microinjections and Stephan Gerlach, Yoshiki Sakai, Alon Oyler-Yaniv, Andrea Herman, Julia Falo-Sanjuan, and Aaron Fultineer for feedback on the manuscript.

## Author contributions

**Conceptualization:** Brandon H. Schlomann, Hernan G. Garcia.

**Data curation:** Brandon H. Schlomann.

**Funding acquisition:** Brandon H. Schlomann, Hernan G. Garcia.

**Investigation:** Brandon H. Schlomann, Ting-Wei Pai, Jazmin Sandhu, Genesis Ferrer Imbert.

**Methodology:** Brandon H. Schlomann.

**Project administration:** Brandon H. Schlomann.

**Resources:** Thomas GW Graham, Hernan G. Garcia.

**Software:** Brandon H. Schlomann.

**Supervision:** Brandon H. Schlomann, Hernan G. Garcia.

**Visualization:** Brandon H. Schlomann.

**Writing – original draft:** Brandon H. Schlomann.

**Writing – review & editing:** Brandon H. Schlomann, Hernan G. Garcia.

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
