## [Decision Letter · Decision Letter 0]

19 Feb 2025

PGENETICS-D-24-01495

Spatial microenvironments tune immune response dynamics in the Drosophila larval fat body

PLOS Genetics

Dear Dr. Schlomann,

Thank you for submitting your manuscript to PLOS Genetics. After careful consideration, we feel that it has merit but does not fully meet PLOS Genetics's publication criteria as it currently stands. Therefore, we invite you to submit a revised version of the manuscript that addresses all the points raised during the review process.

Please submit your revised manuscript within 60 days Apr 20 2025 11:59PM. If you will need more time than this to complete your revisions, please reply to this message or contact the journal office at plosgenetics@plos.org. Please include the following items when submitting your revised manuscript:

We look forward to receiving your revised manuscript.

Kind regards,

Lolitika Mandal, Ph.D

Academic Editor

PLOS Genetics

Aleksandra Trifunovic

Section Editor

PLOS Genetics

Aimée Dudley

Editor-in-Chief

PLOS Genetics

Anne Goriely

Editor-in-Chief

PLOS Genetics

**Journal Requirements:**

https://journals.plos.org/plosgenetics/s/submission-guidelines#loc-parts-of-a-submission

4) We notice that your supplementary Figures are included in the manuscript file. Please remove them and upload them with the file type 'Supporting Information'. Please ensure that each Supporting Information file has a legend listed in the manuscript after the references list.

Potential Copyright Issues:

i) Figures 1A, and 2. Please confirm whether you drew the images / clip-art within the figure panels by hand. If you did not draw the images, please provide (a) a link to the source of the images or icons and their license / terms of use; or (b) written permission from the copyright holder to publish the images or icons under our CC BY 4.0 license. Alternatively, you may replace the images with open source alternatives. See these open source resources you may use to replace images / clip-art:

6) In the online submission form, you indicated that "The raw image data from this manuscript can be obtained from the corresponding author upon request. " All PLOS journals now require all data underlying the findings described in their manuscript to be freely available to other researchers, either

1. In a public repository

2. Within the manuscript itself

3. Uploaded as supplementary information.

7) Please amend your detailed Financial Disclosure statement. This is published with the article. It must therefore be completed in full sentences and contain the exact wording you wish to be published.

8) Please ensure that the funders and grant numbers match between the Financial Disclosure field and the Funding Information tab in your submission form. Note that the funders must be provided in the same order in both places as well. Currently, these funds "Jane Coffin Childs Memorial Fund for Medical Research, the Koret-UC Berkeley-Tel Aviv University Initiative in Computational Biology and Bioinformatics, and a Winkler Scholar Faculty Award" are missing from the Funding Information tab.

**Reviewers' comments:**

Reviewer's Responses to Questions

Reviewer #1: This study investigates the regionalized expression of antimicrobial peptide (AMP) genes along the fat body of Drosophila larvae. This work is particularly significant as the fat body has traditionally been considered a homogeneous organ by most researchers in the field. The authors convincingly demonstrate that the anterior and posterior lobes of the fat body exhibit greater immune reactivity. They further identify genes differentially expressed along the fat body by leveraging a previous microarray dataset. The imaging approach employed here is innovative and provides fresh insights into how spatial microenvironments influence immune responses in Drosophila.

While the study is compelling, it would benefit from a more thorough exploration of the mechanisms underlying the observed heterogeneity. Below, I outline several suggestions to strengthen the manuscript.

Major Recommendations (Easily Addressable)

1. Infection Model Expansion:

It would be valuable to analyze the response of the fat body to oral bacterial infections, such as Erwinia carotovora carotovora 15 (Ecc15) (Basset et al., 2000). Given the proximity of the fat body to the gut, it is worth investigating whether this immune regionalization also occurs in the context of gut-associated infections.

2. Diverse Pathogens:

All results are currently derived using a single bacterial strain, Escherichia coli. To strengthen the findings, I strongly recommend including other bacterial strains such as Pectobacterium carotovorum (Ecc15) or Providencia rettgeri, which produce varying amounts of peptidoglycan and may elicit distinct immune responses.

Recommendations to Further Strengthen the Study

3. Mechanistic Insights into Imd Pathway Regulation:

A plausible hypothesis for the observed differential expression of AMPs is that components of the Imd pathway, such as the PGRP-LC receptor, are unevenly distributed along the fat body.

1. Among the differentially expressed genes identified in the microarray, are there any encoding components of the Imd pathway?

2. GFP-tagged versions of PGRP-LC (Neyen et al., Journal of Immunology, 2012) could be used to investigate whether the receptor's expression varies spatially (but they are weakly expressed).

3. Analyzing Dpt-GFP expression under PGRP-LC overexpression (via heat shock or Gal4/Gal80 systems) could clarify whether the heterogeneity is due to changes in Imd pathway reactivity.

4. Experiments in PGRP-LB mutants (which exhibit reduced peptidoglycan cleavage) could also be informative in determining whether spatial variability persists when the Imd pathway ligand remains uncleaved.

4. Improved Yeast Challenge Model:

The use of Saccharomyces cerevisiae as a challenge model is suboptimal due to its low immunogenicity. Consider using spores from Aspergillus or Beauveria species, which may elicit a more robust immune response.

5. Expression Levels and AMP Dynamics:

The differential expression of AMPs (other than Diptericin) might be partly due to baseline expression levels. For instance, Drosocin and Defensin are expressed at lower levels compared to Drosomycin upon infection. This could be mentionned.

6. Role of Abd-A in Immune Heterogeneity:

Exploring the potential involvement of Abd-A in shaping the spatial heterogeneity of the immune response could provide mechanistic insights on the regionalization.

Minor Recommendation

7. Injection Method Clarification:

The mode of bacterial delivery (microinjection versus pricking) is an important methodological detail that should be explicitly mentioned in the Results section. Additionally, testing whether similar results are observed with pricking (a method that may minimally disrupt larvae) would enhance the robustness of the conclusions.

8. Contextual Background:

Including more information on the development and heterogeneity of the fat body in the Introduction or Discussion sections would provide valuable context for the reader and enhance the manuscript.

Overall Assessment

This manuscript makes a significant contribution to our understanding of immune responses in Drosophila by challenging the traditional view of the fat body as a homogeneous organ. The imaging approach and findings are novel, but incorporating the above suggestions would strengthen the mechanistic depth and applicability of the work.

Reviewer #2: In this manuscript, the authors study the expression pattern of antimicrobial peptides in the fat body of Drosophila larvae injected with bacteria. They use state-of-the-art microscopy techniques and quantification tools. Analysis of the images obtained enabled them to conclude that Dipt A is expressed in a U-shaped expression pattern, with a maximum expression level in the anterior and posterior lobes of the fat body and a minimum in the center. Similar results were obtained using reporters for other AMPs under the control of either the IMD or the Toll pathways. A detailed analysis of the dynamics of AMP expression in cells led the authors to conclude that AMP expression is largely deterministic with spatially varying rates, rather than varying activation delays or stochastic dynamics. By analyzing the position of bacteria in larvae, the authors test the hypothesis that differential concentrations of bacteria in larvae explain the pattern of AMP expression. The results show that local bacterial concentration is not the main determinant of Dipt A expression patterns. Finally, the authors studied the role of the cardiac tube in the movement of bacteria in the larva and therefore potentially in the DiptA expression pattern, by blocking the circulation of hemolymph associated with cardiac movement. These experiments show that while the heart is indeed involved in bacterial movement, it is not involved in the establishment of the AMP expression pattern.

The manuscript is well written, with quality images and adhoc quantification of the phenotype studied.

Major issues

- The main conclusion of this manuscript is that DptA and AMPs in general are preferentially expressed in the anterior and posterior lobes of bacteria-infected larvae. Indeed, the second part of the manuscript, which seeks to determine the reasons for this expression pattern, is inconclusive. In fine, the manuscript remains very descriptive (of a already reported phenotype, see below) and fail short in providing on the mechanisms involved;

- The authors express surprise at this bi-modal expression pattern, which differs from what has been obtained and published with Dipt LacZ. However, it is undoubtedly the latter that does not reflect reality. Indeed, there are already images in the literature showing this U-shaped activation of DiptA in bacterially infected larvae. In PMID: 19482944, figure 4 shows photos of Dipt Cherry larvae infected with Ecc bacteria. The U-shape expression pattern is clearly visible. It is interesting to note that these larvae were infected by ingestion of Ecc bacteria and not by injection as in the present manuscript. However, it is accepted that orally ingested Ecc do not cross the gut epithelium and can induce fat body Dipt expression as a U-shape without being present in the hemolymph. In view of these results, the experiments presented in this manuscript testing the role of bacterial localization are not really relevant, since the same result can be obtained without contact between the bacteria and the fat body. Finally, I'd like to point out to the authors that AMPs can be induced independently of bacteria, by stress, and that this is also reflected in this U-shape expression. This suggests that the anterior and posterior lobes of the fat body have a heightened sensitivity to the rest of the fat body to activate AMPs, which is clearly independent of the local concentration of bacterial elicitor and corresponds rather to an intrinsic characteristic of the tissue potentially linked to their developmental history. It would be interesting to test the role of the hox genes that segment the larva in the acquisition of this characteristic.

Reviewer #3: In this manuscript, Schlomann and colleagues present the intriguing observation that immune response to bacterial injection in the larval fruit fly fat body is spatially-structured. They test several hypotheses about the origins of this structure, concluding that certain regions may be primed to produce antimicrobial peptides.

The observation of the spatially-structured response is interesting and novel. The methods are presented in detail, and there was substantial methods development required for the work, especially for long-term live imaging of these samples. The authors do not have definitive evidence for the priming hypothesis, but the paper provides useful empirical information on the open problem of the role of tissue structure in immune response. However, I do not think that the conclusion that activated regions have higher bacterial exposure due to blood flow is supported by the data, as presented. Additionally, I would like to see stronger analysis to support the claim that the patterning is not due directly to an uneven distribution of bacteria, and more generally on the relationship between bacterial exposure and AMP production.

Comments:

Bacterial concentration, heartbeat, and response: This section is confusing, and I do not believe the conclusions are supported by the data as presented. There are three separate questions here: 1. Does bacterial load in a particular region predict response, 2. Does heartbeat/blood transport determine the spatial distribution of bacteria; and 3. Does heartbeat affect the spatial response pattern. 1 and 3 are questions that bear directly on the causes of the spatial patterning, while 2 is primarily distracting. Further, I do not think that the conclusion the authors draw in answer to 2 is supported by Figure 4. In particular, the authors claim that, due to blood flow, planktonic bacteria occur preferentially at the anterior and posterior. However, from Figure 4, there appears to be barely any additional bacteria in the anterior in the presence of heartbeat; the spatial pattern in fact appears a bit stronger in the absence of heartbeat. Thus it would seem that the heartbeat actually has little effect on the spatial distribution of the bacteria. In fact, it seems like the main effect is that there are substantially more planktonic bacteria everywhere in the absence of heartbeat.

The authors could reorganize this section to clarify it. Additionally, I would like to see clearer analysis to support the claim that bacterial load does not determine the AMP response. This analysis would extend Figure S6, which appears to me to bear more on the central questions here than much of Figure 4. In particular, the authors could analyze the extent to which the observed variation in cellular activation can be explained by different densities of bacteria, given no knowledge of spatial location: how well does the best monotonic model mapping a bacterial density to a response amplitude do in predicting responses? How does this compare to a model based only on physical location? Additionally, given the heterogeneity from larva to larva, the authors should do this analysis at the single-larval level if possible. It would also be useful to see maps of DptA-GFP fluorescence distribution and bacterial distribution for individuals. These analyses would add substantially to demonstrating that bacterial distribution cannot explain the observed spatial patterning.

Additional comments on this section: 1. Does the additional planktonic bacterial load in the absence of heartbeat lead to higher overall DptA-GFP response?

2. I think the presentation would be clearer if this section had a first figure relating bacterial density to DptA-GFP response, and a second figure showing that knocking out heartbeat does not erase the spatial pattern.

Based on these considerations, I do not think that the sentences in the abstract ‘This pattern correlates with microbial localization via blood flow but is not caused by it: loss of heartbeat suppresses microbial transport but leaves the expression pattern unchanged. This result suggests that regions of the tissue most likely to encounter microbes via blood flow are primed to produce antimicrobials’ are supported.

**Have all data underlying the figures and results presented in the manuscript been provided?**

Reviewer #1: Yes

Reviewer #2: Yes

Reviewer #3: Yes

PLOS authors have the option to publish the peer review history of their article (what does this mean?). If published, this will include your full peer review and any attached files.

Reviewer #1: **Yes:** Bruno Lemaitre

Reviewer #2: No

Reviewer #3: No

**Figure resubmission:**
---

## [Decision Letter · Decision Letter 1]

16 Sep 2025

PGENETICS-D-24-01495R1

Spatial microenvironments tune immune response dynamics in the Drosophila larval fat body

PLOS Genetics

Dear Dr. Schlomann,

Thank you for submitting your manuscript to PLOS Genetics. After careful consideration, we feel that it has merit but does not fully meet PLOS Genetics's publication criteria as it currently stands. Therefore, we invite you to submit a revised version of the manuscript that addresses the points raised during the review process.

Please submit your revised manuscript within 30 days Oct 16 2025 11:59PM. If you will need more time than this to complete your revisions, please reply to this message or contact the journal office at plosgenetics@plos.org. Please include the following items when submitting your revised manuscript:

We look forward to receiving your revised manuscript.

Kind regards,

Lolitika Mandal, Ph.D

Academic Editor

PLOS Genetics

Paula Cohen

Section Editor

PLOS Genetics

Aimée Dudley

Editor-in-Chief

PLOS Genetics

Anne Goriely

Editor-in-Chief

PLOS Genetics

**Additional Editor Comments:**

Reviewer #1:

Reviewer #2:

Reviewer #3:

**Journal Requirements:**

**Reviewers' comments:**

Reviewer's Responses to Questions

**Comments to the Authors:**

Reviewer #1: The article has been improved. I recommend its publication. They have addressed most of my recommendation.

Reviewer #2: This manuscript describes a phenomenon already reported whereby, in larvae infected with bacteria, the transcriptional activation of diptericin, a readout of the activation of one of the NF-kB IMD pathways, is uneven and more pronounced in the anterior and posterior ends of the fat body lobes. The added value of this study lies in its quantitative aspect, which provides a more accurate assessment of the phenomenon described. My main criticism was that this study remained highly descriptive, without demonstrated mechanisms to understand the cellular and molecular explanations for this differential expression and its function. In the revised version, the authors show that overexpression of a constitutive version of PGRP-LC mimics the injection of E. coli, suggesting that the central part of the fat body is not competent to activate the IMD pathway. However, we still do not know the reasons for this incompetence. Surprisingly, the mCD8-GFP protein, which has nothing to do with the IMD pathway, also has a U-shaped pattern. Furthermore, the use of E faecalis as an inducer shows that the Toll pathway is activated throughout the fat body. My impression after reading this new version of the manuscript remains the same. Through quantification, they describe a heterogeneity of activation of the IMD pathway in the fat body (already described). There is no data available to enable us to assess which cellular and molecular mechanisms (i) that explain this heterogeneity, (ii) that explain why what is true for the IMD pathway is not for the Toll pathway, and (iii) determine the role of this phenomenon in the immune response or in the biology of the larva more generally.

Reviewer #3: The authors have strengthened and clarified the manuscript; I have no further comments/concerns.

**Have all data underlying the figures and results presented in the manuscript been provided?**

Reviewer #1: Yes

Reviewer #2: Yes

Reviewer #3: Yes

PLOS authors have the option to publish the peer review history of their article (what does this mean?). If published, this will include your full peer review and any attached files.

Reviewer #1: No

Reviewer #2: No

Reviewer #3: No

**Figure resubmission:**
---

## [Decision Letter · Decision Letter 2]

12 Jan 2026

Dear Dr Schlomann,

We are pleased to inform you that your manuscript entitled "Spatial microenvironments tune immune response dynamics in the Drosophila larval fat body" has been editorially accepted for publication in PLOS Genetics. Congratulations!

Yours sincerely,

Paula E. Cohen

Section Editor

PLOS Genetics

Paula Cohen

Section Editor

PLOS Genetics

Aimée Dudley

Editor-in-Chief

PLOS Genetics

Anne Goriely

Editor-in-Chief

PLOS Genetics

BlueSky: @plos.bsky.social

Comments from the reviewers (if applicable):

Reviewer's Responses to Questions

**Comments to the Authors:**

Reviewer #2: The authors have now softened some of their claims. I recommend publication

**Have all data underlying the figures and results presented in the manuscript been provided?**

Reviewer #2: Yes

PLOS authors have the option to publish the peer review history of their article (what does this mean?). If published, this will include your full peer review and any attached files.

Reviewer #2: **Yes:** Julien ROYET

**Data Deposition**

http://datadryad.org/submit?journalID=pgenetics&manu=PGENETICS-D-24-01495R2

**Press Queries**

---

## [Editor Report · Acceptance letter]

PGENETICS-D-24-01495R2

Spatial microenvironments tune immune response dynamics in the Drosophila larval fat body

Dear Dr Schlomann,

We are pleased to inform you that your manuscript entitled "Spatial microenvironments tune immune response dynamics in the Drosophila larval fat body" has been formally accepted for publication in PLOS Genetics! Your manuscript is now with our production department and you will be notified of the publication date in due course.

With kind regards,

Narmatha Raju, M.Sc

PLOS Genetics

On behalf of:
